# HPLC, FTIR and GC-MS Analyses of *Thymus vulgaris* Phytochemicals Executing In Vitro and In Vivo Biological Activities and Effects on *COX-1*, *COX-2* and Gastric Cancer Genes Computationally

**DOI:** 10.3390/molecules27238512

**Published:** 2022-12-03

**Authors:** Ayesha Saleem, Muhammad Afzal, Muhammad Naveed, Syeda Izma Makhdoom, Modasrah Mazhar, Tariq Aziz, Ayaz Ali Khan, Zul Kamal, Muhammad Shahzad, Metab Alharbi, Abdulrahman Alshammari

**Affiliations:** 1Department of Basic and Applied Chemistry, Faculty of Sciences, University of Central Punjab, Lahore 54000, Pakistan; 2Department of Biotechnology, Faculty of Science and Technology, University of Central Punjab, Lahore 54590, Pakistan; 3School of Food & Biological Engineering, Jiangsu University, Zhenjiang 212013, China; 4Department of Biotechnology, University of Malakand, Chakdara 18800, Pakistan; 5Department of Pharmacy, Shaheed Benazir Bhutto University Sheringal, Dir Upper 18000, Pakistan; 6School of Biological Sciences, Health and Life Sciences Building, University of Reading, Reading RG6 6AX, UK; 7Department of Pharmacology and Toxicology, College of Pharmacy, King Saud University, P.O. Box 2455, Riyadh 11451, Saudi Arabia

**Keywords:** *T. vulgaris*, phytoconstituents, docking, anti-inflammatory, drug designing, hepatotoxicity

## Abstract

Medicinal plants have played an essential role in the treatment of various diseases. *Thymus vulgaris*, a medicinal plant, has been extensively used for biological and pharmaceutical potential. The current study was performed to check the biopotential of active biological compounds. The GC-MS analysis identified 31 compounds in methanolic crude extract, among which thymol, carvacrol, p-cymene, and eugenol are the main phytoconstituents present in *T. vulgaris*. The HPLC analysis quantified that flavonoids and phenolic acids are present in a good concentration in the active fraction of ethyl acetate and *n*-butanol. FTIR confirmed the presence of functional groups such as phenols, a carboxylic group, hydroxy group, alcohols, and a benzene ring. Among both fractions, ethyl acetate showed high antioxidant activity in the DPPH (84.1 0.88) and ABTS (87.1 0.89) assays, respectively. The anti-inflammatory activity of the fractions was done in vitro and in vivo by using a carrageenan-induced paw edema assay, while the hexane-based extract showed high anti-inflammatory activity (57.1 0.54) in a dose-response manner. Furthermore, the lead compound responsible for inhibition in the denaturation of proteins is thymol, which exhibits the highest binding affinity with COX1 (−6.4 KJ/mol) and COX2 (−6.3 KJ/mol) inflammatory proteins. The hepatotoxicity analysis showed that plant-based phytoconstituents are safe to use and have no toxicity, with no necrosis, fibrosis, and vacuolar degeneration, even at a high concentration of 800 mg/kg body weight. Furthermore, the in silico analysis of HPLC phytochemical compounds against gastric cancer genes showed that chlorogenic acid exhibited anticancer activity and showed good drug-designing characteristics. Thrombolysis and hemolysis are the major concerns of individuals suffering from gastric cancer. However, the *T. vulgaris* fractions showed thrombolysis from 17.6 to 5.4%; similarly, hemolysis ranged from 9.73 to 7.1% at a concentration of 12 mg/mL. The phytoconstituents present in *T. vulgaris* have the potential for multiple pharmacological applications. This should be further investigated to isolate bioactive compounds that can be used for the treatment of different ailments.

## 1. Introduction

Plants have been used by man throughout history in a variety of ways [1]. In scientific developments, there have been numerous studies on medicinal plants worldwide due to their therapeutic efficacy. Medicinal plants are an important source of lead compounds for developing new drugs that are highly effective, have no side effects, and are economically feasible [2,3,4,5,6,7,8,9,10]. According to studies conducted by the World Health Organization, over 80% of the world’s population relies on medicinal herbs for the treatment of various ailments. Plants have been used by humans for over 60,000 years based on fossil fuel records [11]. Many studies have reported that anti-inflammatory, antibacterial, antifungal, antimalarial, anticancer, antioxidant, and other biological activities depend on a variety of phytoconstituents that are isolated from various medicinal plants [12]. Several exogenous factors, including drugs, chemicals, smoke, and environmental stress conditions, can produce reactive oxygen species (ROS). It has become clear that these factors are an important cause of diseases such as atherosclerosis, inflammation, neurodegenerative diseases, cardiovascular diseases, and cancer [13]. The excess production of free radicals such as OH^+^, O^2−^, etc. in the human body results in the damaging of cells. It causes severe oxidative damage to biological vital molecules such as lipid, deoxyribonucleic acid (DNA), and protein [14]. Phenolic compounds are natural antioxidants that protect against free radical-induced illnesses. A variety of natural antioxidants contain phenolic compounds, such as curcuminoids, phenolics, lignans, tannins, coumarins, and flavonoids. As a matter of fact, antioxidants of natural origin have received considerable attention from the health and food industries in regard to identifying secondary metabolites. Antioxidants protect the body from radical damage by scavenging reactive oxygen species (ROS) [15].

Inflammation is a biological process that is caused due to tissue injury, chemical irritation, and any sort of viral or pathogenic infection. This response is usually triggered by innate immune system receptors because of pain, redness, warmth, and swelling caused by pathogens [16]. It is classified into two types: chronic and acute. Acute inflammation is the body’s response to potential damage. The symptoms include pain, loss of function, and swelling. Prostaglandins, prostacyclins, and thromboxane are all produced in the body by cyclooxygenase (COX), which contributes to pain, inflammation, and platelet aggregation. Several pathological conditions such as cardiovascular disease, inflammation, and cancer have been associated with the overexpression of inflammatory proteins such as COX1, COX2, and proinflammatory metabolites of arachidonic acids [17]. Cancer is the deadliest disease mainly caused due to the uncontrolled growth of cells. The most common type of cancer is cancer of the breasts, lungs, skin, colorectal, prostate, and gastric. Among all types of cancer, gastric cancer is the second most common [18]. The carcinogenesis of gastric cancer refers mainly due to the accumulation of alterations of multiple genes such as tumor suppressor genes, mismatch repair genes, and oncogenes. From the GLOBOCAN 2018 data, gastric cancer has been found in over 1.03 million people, resulting in almost 78,300 deaths. Helicobacter pylori, smoking, obesity, alcohol, and red meat are the main factors for gastric cancer [19].

*Thymus vulgaris* belongs to the Lamiaceae family and is widely used in folk medicines. It is mostly found in the Southern Europe Mediterranean, North Africa, and Asia regions with approximately 300 species [20]. *Thymus vulgaris* is the source of many phytoconstituents (secondary metabolites), such as phenols, tannins, glycosides, and flavonoids, which are responsible for many biological activities [21]. The current study determines the phytochemical constituents through GC-MS and HPLC analyses of two fractions: ethyl acetate and *n*-butanol from *T. vulgaris* methanolic extract. The extracted plant was analyzed for various in vitro and in vivo biological assays, i.e., anti-inflammatory, antioxidant, hemolytic, thrombolytic, pyretic, and antidiabetic activities, along with a computational analysis on gastric cancer-causing genes by quantified phytochemicals. The present research reveals the potent phytochemicals that can be a benchmark in the cure of various ailments.

## 2. Results

### 2.1. Phytochemical Screening

Flavonoids and phenolics are the most important group of plants and exhibit strong antioxidant potential. The phytochemical screening of *n*-butanol and ethyl acetate fractions of *T. vulgaris* depicted the flavonoids, saponins, alkaloids, terpenoids, tannins, glycosides, steroids, and phenols. Alkaloids are absent in both fractions, while phenols, flavonoids, and terpenoids are present in both fractions, whereas saponins and tannins are absent in both fractions. and steroids and glycosides are absent in ethyl acetate.

### 2.2. Gas Chromatography-Mass Spectrometry

The chemical composition of methanolic plant extract was determined through a GC-MS analysis, as shown in Table 1 and Figure 1. The major compounds were thymol (28.88), p-cymene (7.77), eugenol (4.66), and γ-terpinene (3.47), which suggests that these fractions of *T. vulgaris* belong to the thymol chemotype (Table 1), in agreement with a study reported by Romania.

### 2.3. Total Phenolic Content

The fractions of *T. vulgaris* were found to contain the total phenolic content (TPC). Ethyl acetate had the highest TPC value 148.3 ± 1.54 mg/g of dry weight GAE equivalent, whereas *n*-butanol had the lowest value of 126.69 ± 1.37 mg/g of dry weight GAE equivalent.

### 2.4. Total Flavonoid Content

The flavonoid content in the *T. vulgaris* fractions of ethyl acetate and *n*-butanol were represented as mg/g of dry weight of quercetin equivalents. Ethyl acetate had the highest TFC (108.9 ± 0.92 mg/g), followed by *n*-butanol (74.1 ± 0.81 mg/g).

### 2.5. High-Performance Liquid Chromatography

The quantitative analysis of the ethyl acetate and *n*-butanol fractions is presented in Table 2. The components of chlorogenic acid, caffeic acid, benzoic acid, sinapic acid, gallic acid, kaempferol, myricetin, and quercetin were identified in fractions of *T. vulgaris* in comparison to the retention time. The results of the HPLC analysis showed that the ethyl acetate fraction contains the highest concentration of phytochemicals (chlorogenic acid 0.012, caffeic acid 0.151, sinapic acid 0.115, benzoic acid 0.343, and vanillic acid 0.144 mg/g) than *n*-butanol (chlorogenic acid 0.044, caffeic acid 0.154, and sinapic acid 0.046). The flavonoid analysis of the *T. vulgaris* ethyl acetate and *n*-butanol fractions showed that chloroform contains a high concentration of quercetin and kaempferol at 0.811 and 0.204 and, in *n*-butanol, 0.473 and 0.109 mg/g, respectively. The results showed that quercetin is the most dominant flavonoid in the *T. vulgaris* HPLC analysis, followed by myricetin (0.380 mg/g) in hexane (Figure 2).

### 2.6. Fourier-Transform Infrared Spectroscopy (FTIR)

The methanolic extract of *T. vulgaris* was shown to have several bioactive components that were confirmed by FTIR spectroscopy, including alcohol, carboxylic acid, phenolic acids, and aromatic compounds. Seven functional groups identified from the methanolic extract are shown in Table 3 and Figure 3. The powerful instance peak is located at peak number 3339, which assigns the O-H stretch.

### 2.7. Antioxidant Activity (DPPH and ABTS Assays)

The antioxidant activity of *T. vulgaris* fractions was assessed by DPPH and ABTS. The results showed that the antioxidant activity was concentration dependent. Among various methods, a high percentage of inhibition was observed by ethyl acetate in ABTS (87.1 ± 0.88), followed by 84.1 ± 0.8 as compared to chloroform, which confirmed the high antioxidant potential of ethyl acetate, which was close to the standard ascorbic acid, as shown in Figure 4. The ability to scavenge free radicals from *Thymus Vulgaris* varies significantly (*p* < 0.05).

### 2.8. Antidiabetic Activity (Starch Iodine and DNSA Assays)

The *T. vulgaris* fractions were concentration-dependent on the inhibition of α-amylase (Figure 5). Among both fractions, ethyl acetate showed the maximum inhibition in the DNSA (89.4 ± 0.95) (Figure 5A) and starch iodine (86.5 ± 0.89) (Figure 5B) assays that showed potential antidiabetic activity as compared to butanol. As the concentration decreased, the antidiabetic activity also decreased and was the minimum at the lowest concentration.

### 2.9. Anti-Inflammatory Activity In Vitro

The ability of *T. vulgaris* fraction’s anti-inflammatory mechanism to prevent the denaturation of bovine serum albumin was calculated at various concentrations (Figure 6). It was shown that the ethyl acetate fractions inhibit the denaturation of BSA less effectively than standard diclofenac. The findings indicated that *T. vulgaris* inhibits protein denaturation in a concentration-dependent manner. *n*-Butanol showed the lowest inflammation at 64.8 ± 0.61, while the ethyl acetate fraction showed a high percentage inhibition at 67.7 ± 0.68, respectively. The ability of protein inhibition in the *T. vulgaris* fractions was significant at *p* > 0.05.

### 2.10. Anti-Inflammatory Activity In Vivo

Figure 7 shows the outcomes of testing the anti-inflammatory activity of *T. vulgaris* fractions in vivo by the carrageenan-induced paw edema method and the results. In the control group, the paw size increases gradually, and swelling in the control group increases due to the untreated nature. The pretreated rats with fractions of *T. vulgaris* were found to be more significant in the reduction of the paw volume at the 4th hour. The ethyl acetate fraction showed the highest percentage of inhibition at 57.1 ± 0.59 at the 400-mg/kg dose, which is close to the standard diclofenac at 60.6 ± 0.7% at a 400-mg/kg dose, as shown in Figure 6. All values of the fractions of *T. vulgaris* are significant at *p* > 0.05.

### 2.11. Effect of GC-MS Quantified Phytoconstituents on COX1 and COX2 Genes

#### Molecular Docking

AutoDock Vina was used to determine the binding energies, and the highest binding scores obtained by thymol, carvacrol, *p*-Cymene, and eugenol were −6.4 KJ/mol, −6.3 KJ/mol, −6.3 KJ/mol, and −6.2 KJ/mol in COX1 and −6.3 KJ/mol, −6.2 KJ/mol, −6.2 KJ/mol, and −6.1 KJ/mol in COX2 inflammatory proteins, while phytol and neophytadiene had the lowest scores of −5.1 and −4.4 and −5.5 and −5.1. It shows that thymol and carvacrol could be the candidates for the anti-inflammatory agent, as both had the highest binding scores. The alkyl, hydrogen bonds, and van der Walls forces in the assessed compounds, as well as pi-interactions between the ligand and inflammatory protein, were the main causes of binding interactions of the compounds. The activity and stability of the substances and proteins may be owed to interactions between the hydrophobic residues and COX1 and COX2, such as pi–alkyl, pi–pi, pi–stigma, van der Walls, and alkyl bonds (Figure 8).

### 2.12. Pharmacokinetic Study (ADME)

The Swiss ADME analysis revealed that thymol, carvacrol, p-cymene, and eugenol were the best anti-inflammatory compounds, as they had lipophilicity Log p less than 4, showing good water solubility, except phytol and neophytadiene with Log p values greater than 4, which were 4.85 and 5.05, as they were poor and moderately soluble in Log S (Ali), Log S (ESOL), and Log S (SILICOS-IT), as shown in Appendix A. The pharmacokinetic analysis of all anti-inflammatory compounds revealed good gastrointestinal absorption and did not violate the Lipinski, Egan, and Veber rules, except for phytol and neophytadiene, and all compounds violated the Ghose and Muegge rules, except for eugenol and spathulenol because of molecular weights greater than 160 g/mol. The pharmacokinetics analysis and interaction of the compounds with cytochrome P450A and P-glycoprotein revealed that all anti-inflammatory compounds showed high gastrointestinal absorption with low glycoprotein permeability and no inhibitory effect on CYP2C19 and CYP2C9, except for phytol and neophytadiene. Appendix A in the Appendix A shows a radar plot, which indicates that all the compounds were within the ranges of the physicochemical properties and that these compounds are suitable for oral administration. The results of these calculations largely correspond to their observed lipophilicity and water-solubility.

### 2.13. Hepatotoxicity

The results revealed that the ethyl acetate and *n*-butanol fractions of *T. vulgaris* were safe for oral administration in acute doses of 200, 400, 600, and 800 mL/kg body weight and showed no sign of toxicity. The liver function parameters were used for liver function tests, and according to the results, the plant fraction slightly reduced the ALT, AST, and ALP activities of ethyl acetate and *n*-butanol-induced hepatoxicity rats, as shown in Table 4. On the AST, rat groups 2–5 treated with the *T. vulgaris* ethyl acetate and *n*-butanol fractions with four concentrations recorded the AST activity close to that of the control. The total protein and globulin levels gradually increased from concentrations of 200–800 mg/kg body weight in both fractions.

A micrograph examination showed that ethyl acetate concentrations from 200 to 400 mg/kg showed slight fibrosis, with no ballooning, nuclear degeneration, and no accumulation of immune cells, and, at high concentrations of 600–800 mg/kg body weight, the results showed slight portal activity, no vacuolar degeneration, a slight infiltration of macrophages, no nuclear variation, and hydropic degeneration. In the case of *n*-butanol fractions, at lower concentrations, there was a slight ballooning of hepatocytes and no accumulation of ECM. Overall, the liver tissue seems healthy and, at high concentrations of 600–800 mg/kg body weight, showed slightly ballooning, and no accumulation of ECM was observed. Further examination showed no nuclear variation or vacuolar degeneration, as shown in Figure 9 and Figure 10.

### 2.14. Effect of HPLC Identified Phytoconstituents on Gastric Cancer Genes

#### 2.14.1. Potential Disease Target Genes

The target genes related to gastric cancer were searched in GeneCards, which includes 3768 genes in GeneCards, with no overlapping target gene. A total of 23 gastric cancer genes were identified from GeneCards. The genes were selected based on previous studies which were important from a drug designing point of view. The complete gene sequences of 23 Gastric cancer genes were retrieved from the PDB with the subsequent PDB ID MULT (IBKN), CTTNA (1DOV), CDKNA (1G3N), MUTHY (1RRS), TRET (2B2A), PIK3A (2ENQ), APC (1DEB), BRCA2 (8HQU), TP53 (1A1U), PTEN (7JVX), ERBB2 (50B4), EGFR (2N5S), POLE (5VBN), RAD51D (2KZ3), SMAD4 (1YGS), FGFR2 (1DJS), DICER-1 (2EB1), THBS2 (1YO8), ABCG2 (5NJ3), MET (5LSP), BRAF (1UWH), CDH1 (2O72) and COL1A1 (1Q7D). Visualization of genes was done through Discovery Studio Visualizer and the complex was made by PyMol.

#### 2.14.2. Structural Retrieval of Phytochemicals (PubChem)

Structures of phytochemicals identified through HPLC analysis are chlorogenic acid (1,794,427), sinapic acid (637,775), benzoic acid (243), gallic acid (370), and caffeic acid (689,043) retrieved through PubChem and saved in the form of PDB.

#### 2.14.3. Venn Analysis (Bioinformatics and Evolutionary Genomics System)

In the Venn diagram intersection of recognized targets about identified chemical compounds and gastric cancer (Figure 11), a total of 23 gastric cancer genes are acquired among which the top six genes ABCG2, MUTHY, TRET, POLE, BRAF and G, and FGFR2 were used to produce Venn diagram.

#### 2.14.4. Protein–Protein Network Construction (STRING)

PPI networks are based on a confidence level greater than 0.40 and hiding nodes associated with independent targets. In PPI, proteins are represented by the nodes and protein-protein interactions by the edges. There are 19 nodes in the network, 81 edges, and 8.53 average degrees between nodes (Figure 12).

### 2.15. Molecular Docking Analysis

The molecular docking of five compounds was performed with all targeted genes and docking scores. Among all compounds, chlorogenic acid shows high docking energy with ABCG2 and Muthy being −8.8 and −8, respectively (Figure 13).

#### 2.15.1. Molecular Dynamic Simulation

Less distortion is visible at the position of each residue capacity level in the resulting model (Figure 14). The Muthy and chlorogenic acid complex has an Eigon value of 6.29415 × 10^−5^. The Figure discusses the specific outcomes of the molecular dynamic simulation in detail. The strongly co-related regions in the heat maps and low RMSD suggest improved interactions of residues. 

#### 2.15.2. Cloud 3D-QSAR Modelling (3-D QSAR)

The in silico 3D-QSAR study was done to investigate the effect of structural characteristics of targeted compounds on biological activities. The method mainly uses three-dimensional properties to predict the biological activities of the ligands through chemometric techniques namely called ANN, GFA, PLS and MLR, etc. A data set of 5 phytochemicals obtained through HPLC analysis was used to build the QSAR model. The generated model was validated by predicting the activity of the top best ligand. To validate the model, the ligand with the best activity was predicted. A good statistic was obtained for chlorogenic acid among all the models. There was a significant r^2^ = 1 and cross-validated correlation coefficient q^2^ −0.1072. The contour map of the best hit compound is shown in Figure 15.

#### 2.15.3. Pharmacokinetic ADME Evaluation

SwissADME analysis showed that chlorogenic acid is the best anti-cancer compound against gastric cancer genes as it has good lipophilicity Lop p less than 4 Log *P*_o/w_ (0.87) and shows good water solubility. Physiochemical properties show molecular formula C16H18O9 and 354.31 g/mol molecular weight. The pharmacokinetic analysis revealed good GI absorption and does not violate Lipinski’s rule. Compound interaction and pharmacokinetic analysis revealed low glycoprotein permeability and no inhibitory effect on CYP2C9 and CYP2C19. BOILED-Egg image of chlorogenic acid is shown in Figure 16.

#### 2.15.4. Cytotoxicity Analysis (CLC Pred)

In addition, CLC-Pred analysis showed chlorogenic acid possesses potent anticancer activity against various cancer and non-cancer cell lines (Table 5). The Pa value of chlorogenic acid against various cancer cell lines is found to be in the range of 0.581–0.023, respectively. Chlorogenic acid is observed to be most effective against lung cancer, liver cancer, breast cancer, metastatic melanoma, and pancreatic carcinoma. In the non-cancer cell line, it is effective against embryonic lung fibroblast.

### 2.16. Hemolytic Activity

*T. vulgaris* fractions at different concentrations and 0.1% Triton-X were screened against human blood cells for hemolytic activity as shown in Table 6. Results show that the highest hemolytic activity was exhibited by n-*n*-butanol fraction 8.04 ± 0.08 and the lowest by ethyl acetate 10.6 ± 0.07 at 12 mg/mL concentration. Lysis of erythrocytes was found to be decreased with decreasing concentrations of fractions. The ability of hemolysis varies significantly (*p* < 0.05).

### 2.17. Thrombolytic Potential

The thrombolytic activity of *Thymus vulgaris* fractions was observed ranging from 9.6 ± 0.25 to 17.6 ± 0.32%. The maximum thrombolytic activity was observed by *n*-butanol at 15.4 ± 0.28 while the minimum was shown by ethyl acetate at 12.3 ± 0.43 (Figure 17).

### 2.18. Anti-Pyretic Activity

The antipyretic activity of the fractions of *T. vulgaris* shows significant results (Figure 18) (*p* > 0.05) at a dose of 200 mg/kg and 400 mg/kg. the highest percentage reduction is seen in ethyl acetate while the lowest is in *n*-butanol fraction as shown in Figure 18.

## 3. Discussion

The phytochemical screening of ethyl acetate and *n*-butanol fractions of methanolic extract of *T. vulgaris* showed the presence of flavonoids. Alkaloids, tannins, terpenoids, glycosides, steroids, saponins and phenols. The presence of these phytoconstituents might contribute to the therapeutic potential of *T. vulgaris* fractions. Biological active compounds were identified through GC-MS and HPLC analysis. The GC-MS analysis of the crude methanolic extract is presented in Table 1. Thirty-one compounds representing 99.97 of the total detected compounds were identified. The major compounds were thymol (28.88%), *p*-cymene (6.68%), carvacrol (7.77%), eugenol (4.66%) and γ-terpinene (3.47%), which suggests the extract analyzed from *T. vulgaris* belongs to the thymol chemotype. A study by Boruga et al. [21] conducted a GC-MS analysis of *T. vulagris* essential oil and reported that thymol (47.59%), γ-terpinene (30.90%) and p-cymene (8.41%) is a major component in oil [21]. Previous studies of *T. vulgaris* extract indicate the presence of various phytoconstituents such as flavonoids and phenols [22]. Ethyl acetate showed the highest total phenolic and flavonoid content 148.3 ± 1.32 and 108.9 ± 1.43, respectively. Previous data revealed that TPC content was 356 ± 9.79 mg eq caffeic acid/g and TFC 186.93 ± 25.19 mg eq rutin/g of *T. vulgaris* [23]. The HPLC analysis of fractions of ethyl acetate and *n*-butanol determines that ethyl acetate contains a high concentration of phenols as caffeic acid, chlorogenic acid, benzoic acid, sinapic acid and vanillic acid as well as flavonoids myricetin 0.204 and quercetin 0.811. FTIR analysis further confirmed the presence of alcohols, carboxylic acids, hydroxy group, aromatic compounds, and saturated aliphatic compounds which supports the biologically active compounds identified through GC-MS, HPLC and phytochemical screening. According to the study by Gedikoglu et al. [24] benzoic acid was found to be the main phenolic compound in HPLC, and quercetin was the main flavonoid in thyme extract of 80% ethanolic and methanolic [25].

Phytoconstituents such as phenolics and flavonoids attributed many of the antioxidant properties to the compounds due to their hydrogen donating ability as well as their structural requirement taken as essential for their strong radical scavenging activity [26]. Several studies revealed that the flavonoid group contained a wide range of biological activities such as antimicrobial, antioxidant, anti-inflammatory, anticancer and anti-allergic [26]. Tannins and its derivative are phenolic compounds that are responsible for antioxidant activity, saponins are involved in resistance to plant diseases because of their role in antimicrobial activity, anti-cancer and anti-diabetic activity and alkaloids contribute to antimicrobial and analgesic activity [27]. The fractions of *T. vulgaris* methanolic extract were identified to have significant antioxidant potential through DPPH and ABTS assay. ABTS method shows higher values as compared to the DPPH method. The strong antioxidant activity in the DPPH method is observed in ethyl acetate fraction 84.1 ± 0.8 and ABTS method 87.1 ± 0.88 as compared to *n*-butanol. Quantitative and qualitative analyses of fraction show that ethyl acetate contains a high number of polyphenols which indicates a direct link of the scavenging activity with the polyphenolic content of the fractions. Our results are supported by a study conducted by Grigore et al. [28] in which antioxidant activity of extract and quercetin showed significant radical scavenging activity at a dose higher than 3 mg/mL exhibiting over 50% of inhibition [29]. The phenolics and flavonoids are responsible for anti-diabetic activity as present results of *T. vulgaris* of different fractions show that the highest enzyme inhibition is obtained in ethyl acetate fraction by the DNSA method. It revealed that ethyl acetate fraction exhibits strong antioxidant, antibacterial and anti-diabetic activity as previous data show a strong association of phenolics and flavonoids with biological activities as they are strong antioxidants [29,30].

Thymol, the primary component of *T. vulgaris* is primarily responsible for its anti-inflammatory properties [31,32]. Present results show the best anti-inflammatory activity (inhibition of BSA protein) in ethyl acetate fraction 64.8 ± 0.61 while the lowest anti-inflammatory activity is shown by *n*-butanol fraction. Albino rats’ paw edema caused by carrageenan was used to assess the in vivo anti-inflammatory action and diclofenac sodium was used as a standard for resulting inhibition measurement. The finding showed that ethyl acetate has stronger anti-inflammatory properties. Our results in similar to an in vivo study conducted by Abdelli et al. [17] in which anti-inflammatory activity of *T. vulgaris* was observed from Tlemcen and Mostaganem and results showed it exhibits strong anti-inflammatory activity of 58.4% and 50.4% as the major compound identified through GC-MS was the thymol [17]. The in silico analysis of inflammatory proteins COX1 and COX2 against bioactive compounds also shows that thymol has a high binding affinity with both proteins and ADMET analysis shows good pharmacokinetics and drug-likeness properties. Thymol inhibits elastase production by activating neutrophils, a marker of inflammation [33] and inhibits COX [34]. The anti-inflammatory properties of thymol are also contributed by its strong antioxidant activity. Aside from that, minor compounds, such as linalool and p-Cymene also contain anti-inflammatory properties as p-cymene inhibits NF-κB and MAPK signaling pathways, thus reducing TNF-α and IL-1β secretion [35].

Phytoconstituent such as alkaloids, phenols, flavonoids, and tannins have been known as hepatoprotective plants. ROS- induced hepatotoxicity can be effectively controlled through the administration of agents possessing antioxidant, free radical scavenger, and anti-lipid per oxidant activities [36,37]. The histopathological study showed healthy cells were observed with normal shape cells and no necrosis while fraction treated cells of both fractions of *T. vulgaris* showed minor changes in liver tissue pattern compared to the control group even at higher concentrations shows no necrosis, fibrosis, and nuclear degeneration. ALT. AST and ALP levels decrease compared to the control group, and total protein and globulin levels increase as concentration increases. This revealed that *T. vulgaris* plant shows no toxicity at a high dose of 800 mg/kg body weight and is safe for oral administration of drugs. The literature review supports these results, but this necrosis was mild and reversible [36]. Furthermore, computational modeling was performed on gastric cancer genes against phytochemicals to investigate the anticancer effect.

In the current work, 23 total target genes for gastric cancer were collected from GeneCards, and a total of five compounds: -chlorogenic acid, gallic acid, sinapic acid, benzoic acid, vanillic acid, and caffeic acid were used as ligands. Chlorogenic acid demonstrated the highest binding affinity to all the target genes among the five ligands. FGFR2, Muthy, ABCG2 and TRET are four of the top 19 target genes with the highest connections in the PPI network that are important in the fight against gastric cancer. according to the molecular docking analysis, Muthy and FGFR2 were also discovered to have higher binding affinity. Nevertheless, there are no pertinent research its impact on stomach cancer. Docking results showed that chlorogenic acid has the highest binding activity with all the targeted genes and could be used as a drug candidate against gastric cancer. The 3D-QSAR analysis also shows that chlorogenic acid exhibits the best q^2^ and r^2^ values. It was also confirmed through ADMET analysis that chlorogenic acid has good lipophilicity and water solubility characteristics, follows all five Lipinski rules, and could be administered orally.

In addition, CLC-Pred analysis showed chlorogenic acid possesses potent anticancer activity against various cancer and non-cancer cell lines. The Pa value of chlorogenic acid against various cancer cell lines is found to be in the range of 0.581–0.250, respectively. Chlorogenic acid is observed to be most effective against lung cancer, liver cancer, breast cancer, metastatic melanoma, and pancreatic carcinoma. In the non-cancer cell line, it is effective against embryonic lung fibroblast. Chlorogenic acid has also been studied as a potential treatment for fibrosis and cancer in several experimental studies [38]. Numerous issues that could arise from utilizing medicinal herbs can be found by toxicological studies, especially in vulnerably disposed of individuals. The potential negative effects of administration remedies using plant extracts must be carefully watched for. This is due to the fact that conventional trails are not particularly helpful when it comes to the issue of risk assessment. Many natural products cause side effects such as hemolytic anemia as some phytochemicals disrupt the erythrocyte cell membrane. Hence most herbs and plant species need to be evaluated for their possible side effects and hemolytic effects. Hemolytic activity should be in the safe range is less than 10% [39]. These gastric cancer-causing genes also cause hemolysis, thrombosis, and fever as a symptom of cancer and it was evaluated in vitro that phytochemical compounds present in *T. vulgaris* exhibited an effect against these causes. As hemolytic results show that the highest hemolysis is shown by an *n*-butanol fraction (8.04 ± 0.08) while the lowest is by an ethyl acetate fraction (7.1 ± 0.21). It shows that all the fractions are within the safe range at high doses that are less than 10% and decrease with decreasing concentrations. The thrombolytic activity and antipyretic activity also confirmed that these fractions have therapeutic potential against diseases as thrombolytic activity of ethyl acetate and *n*-butanol fractions ranged from 12.3 to 15.4% and antipyretic activity ranged from 42.1 to 81.5% at 400 mg/kg dose in *n*-butanol and ethyl acetate, respectively.

## 4. Materials and Methods

### 4.1. Plant Collection and Preparation

The fresh plant of *T. vulgaris* was purchased from Islamabad Nursery, Islamabad, and confirmed by the Botany Department of the University of Central Punjab. The whole plant of *T. vulgaris* was washed with fresh water and dried at room temperature for three days. After drying, 500 g of the whole plant was ground to a fine powder. *T. vulgaris* powder was packed in sealed plastic bottles until they were extracted [40]. 

### 4.2. Extraction Method

Dried whole plant powder of *T. vulgaris* (500 g) was macerated with methanol (1500 mL) for seven days using a Soxhlet extractor to obtain the methanolic extract. In the process of extracting and evaporating, rotary evaporators were used, and solid masses were formed that were amorphous. Water (75 mL) was used to defeat the crude extract (49.12 g). For fractions, a separatory funnel was used to transfer them to different solvents, namely ethyl acetate (0.53 g) and *n*-butanol (0.41 g) yields, respectively [41].

### 4.3. Phytochemical Screening Tests

The phytochemical screening of ethyl acetate and *n*-butanol fractions of *T. vulgaris* was performed by using reported established protocols, respectively [42,43].

### 4.4. Gas Chromatography-Mass Spectrometry (GC-MS)

A gas chromatography-mass spectrometry (GC-MS) analysis of the methanolic extract of the *T. vulgaris* was carried out according to the protocols. A DB-5MS column with 0.25-μm film thickness, 0.25 mm in diameter, and 30 m in length was used to observe the methanolic extract using GC-MS model 7890B, 5977A working at 75 eV of the ionization energy. Helium was used as a carrier gas, which was used at a rate of 1 ml/min. We set the MS transfer line temperature to 280 °C, the split ratio to 1:6, injected 1 μL of the sample, and used a 30-atomic mass unit mass scan. The columns were initially heated to 50 °C for one minute. A temperature rise of 8 °C per minute was set to reach 290 °C with regular intervals of time. With a 1-mL/min flow rate, helium was used as a carrier gas to move 1 mL of sample extract down the column. FID spectroscopy was used to identify and further analyze the components after separation in the column at 75 eV. To identify the molecular weight, name, and chemical structure of these compounds, we used NIST MS 2.0 libraries.

### 4.5. Quantitative Analysis

#### 4.5.1. Total Phenolic Content (TPCs)

Alu’datt et al. [44] reported the Folin–Ciocalteu reagent procedure in 2019 for the estimation of the total phenolic content (TPC), and 0.5 mL of *T. vulgaris* fractions ethyl acetate and *n*-butanol were mixed with 2.5 mL of Folin–Ciocalteu reagent. The test tubes were incubated for 15 min at 37 °C, and 2 mL of Na_2_CO_3_ (7.5% *w*/*v*) solution was added to the mixtures and made up to 10-mL volume by distilled water and was again incubated for 30 min. Then, the absorbance was spectrophotometrically recorded in triplicate at 760 nm. For the reference curve, gallic acid as a standard was used for TPC determination from concentrations 50–250 μg/mL, and values of the total phenols were expressed as gallic acid equivalents GAE mg/g of dry weight by using the equation y = 0.0037x−0.094, R^2^ = 0.999 [44].
**T** = **C** × **V/M**
where **T** = total phenolic content mg/g extract in gallic acid equivalent, **C** = concentration of fractions established from the gallic acid calibration curve, **V** = volume of extract, milliliter (mL), and **M** = weight of dry plant extract, grams (g).

#### 4.5.2. Total Flavonoid Content (TFC)

The total flavonoid content was identified using a protocol reported by Zengin et al. [45] by utilizing the aluminum chloride colorimetric method. Quercetin was used as a reference and was dissolved in methanol to create the standard curve at concentrations of 50–250 μg/mL. In a flask, 0.5 mL of plant fraction solution was combined with 0.1 mL of 10% *w*/*v* solution of aluminum chloride, 0.1 mL of a 10% *w*/*v* solution of aluminum chloride, and 0.1 mL of a 0.1 mM potassium acetates solution, and the volume was kept up to 5 mL by adding distilled water. For 30 min, the solution was held at 37 °C. The results of the triplicated estimates of the TPC were averaged. The absorbance was recorded by using a UV–Vis spectrophotometer at 715 nm. The results were obtained in μg of quercetin equivalent (QE) per mg/g of the extract by equation y = 0.0042x − 0.1149, R^2^ = 0.998 [45]. The value of the flavonoids was calculated according to the formula below:**T** = **C** × **V/M**
where **T** = total flavonoid content mg/g extract in quercetin equivalent, **C** = concentration of fractions established from the quercetin calibration curve, **V** = volume of extract, milliliter (mL), and **M** = weight of dry plant extract, grams (g).

### 4.6. High-Performance Liquid Chromatography (HPLC)

For a quantitative analysis of the phenolics and flavonoids, high-performance liquid chromatography (HPLC) of the plant extract fractions was used. The column used was a shim-packed CLC ODS C-18 (2.5 cm, 4.6 mm, 5 μm in diameter). In their respective solvents, *n*-butanol and ethyl acetate plant extract fractions containing 10 mg/mL were prepared. With a flow rate of 1 mL/min, 20 μL of fractions were mixed with mobile phase A (H_2_O:acetoacetate 94:6, pH 2.27) and B (ACN 100%), which had different parameters at 15% B for 0 min, changed into 45% B for 15–30 min and 100% B at 45% for 35–40 min. UV–visible detector spectra of all samples were recorded at 280 nm [46].

### 4.7. Fourier-Transform Infrared Spectroscopy (FTIR)

The extracts of methanolic extract *T. vulgaris* were mixed with KBr salt, using a mortar and pestle, and compressed into a thin pellet. For spectroscopy measurements of the samples, a Perkin Elmer Spectrum two FTIR spectrometer was used, and the scan range was between 4000 and 500 cm^−1^ [47].

### 4.8. Biological Activities

#### 4.8.1. Antioxidant Activity (DPPH)

The radical scavenging activity of 2,2-diphenyl-1-picrylhydrazyl (DPPH) was assessed by the protocol given by Anwar et al. [28] with modifications. In the dark, 1 mL of 90 μM DPPH was mixed with 1 mL of different concentrations of fractions and ascorbic acid as the standard (12, 6, 3, 1.5, 0.75, and 0.37 mg/mL). Thirty minutes were spent incubating the mixture at 37 °C. An ELISA plate reader was used for the absorbance at 630 nm, and the activity was expressed as a percentage inhibition [48].
**% Radical scavenging activity** = Abs of control − Abs of sample/Abs of control × 100

#### 4.8.2. Antioxidant Activity (ABTS assay)

The radical scavenging method detected by the ABTS method was conducted by using the protocol given by Asif et al. [49]. The stock solution was made by taking an equal quantity of ABTS and potassium per sulphate in a flask and adding up to 40 mL of distilled water, and 100 μL of this suspension was mixed with 39 mL of methanol. After this, 200 μL of plant extract was mixed with 2400 μL of ABTS suspension and kept at room temperature for 15 min. Absorbance is taken from the 96-well ELISA plate reader at 630 nm. The results were calculated and termed in percentages, which were compared with the standard gallic acid [49].
**% Radical scavenging activity** = Abs control − Abs sample/Abs control × 100

#### 4.8.3. Antidiabetic Activity (DNSA method)

The protocol given by Dessalegn et al. [50] was used for the antidiabetic activity by the DNSA assay, and 200 μL of ethyl acetate and *n*-butanol fractions were mixed with 200 μL of α-amylase solution. The reaction mixture was incubated for 10 min at 25 °C. Following the incubation, 200 μL of 1% starch solution were added and further incubated for 10 min. After this, 400 μL of DNSA solution were added, and the absorbance was checked by ELISA at 630 nm. Distilled water was used as a negative control, and metformin as the standard [50].
**% Inhibition** = Abs control − Abs sample/Abs control × 100.

#### 4.8.4. Antidiabetic Method (Starch Iodine Method)

We checked the a-amylase inhibition of different fractions of *T. vulgaris* at different concentrations by using the starch iodine method by the protocols, and metformin was taken as a positive control, with 195 μL of plant fractions and metformin incubated with 5 μL of α-amylase for 10 min at 37 °C. After incubation, 50 μL of the starch solution was added and incubated further for 60 min. Lastly, 100 μL of 1% of iodine solution was added, and the absorbance was measured by using an ELISA reader at 630 nm. Distilled water was taken as a negative control [51].
**% Inhibition** = (Abs Control-Abs sample)/Abs control × 100

#### 4.8.5. Anti-Inflammatory Activity In Vitro

The approach described by Williams et al. [52] was modified to test the impact of the anti-inflammatory activities. Different concentrations of fractions 12, 6, 3, 1.5, 0.75, and 0.37 mg/mL were prepared, and 100 μL of sample were taken inside centrifuged tubes, and 0.5 mL of bovine serum albumin BSA was added. The mixture was incubated at 37° C for 20 min and then kept in a water bath for 10 min at 70 °C. The tubes were cooled down, and absorbance was checked in an ELISA reader at 630 nm in ELISA plates. The percentage inhibition was checked by the given formula [52].
**% BSA inhibition** = (Abs Control − Abs sample/Abs control) × 100

#### 4.8.6. Anti-Inflammatory Activity In Vivo

A carrageenan-induced anti-inflammatory hind paw edema effect was observed. Thirty-six albino male rats weighed in the range of 100–150 g were taken and divided into six groups for each fraction of ethyl acetate and *n*-butanol, each group containing three rats. Every group administrated with drug dosage intramuscularly was as follows: group 1 control; groups 2 to 5 plant extracts at concentrations of 50, 100, 200, and 400 mg/kg of both the fractions; and group 6 diclofenac 10 mg/kg.

Before one hour of dose administration, 0.1 mL of 1% carrageenan solution in normal saline was given to the left-hand paw under sub-plantar tissue for 1 h to induce inflammation, and the right-hand paw served as a control. The volume/diameter of the paw were measured at 0, 60, 120, and 180 min by using a digital vernier caliper to check the increase in paw edema. An increase in volume was observed as an indication of inflammation showing edema. The following formula was used to calculate the anti-inflammatory activity of the extracts and standard [53]:**% Paw edema** = Io − It/Io × 100
where Io shows the paw mean size of a control group, and it shows the paw mean size of the treated groups.

### 4.9. Effect of GC-MS Quantified Phytoconstituents on COX1 and COX2 Genes

#### 4.9.1. Molecular Docking

Docking was performed by using AutoDock Vina. The 3D structures of COX1 (PDB: 6Y3C) and COX2 (PDB: 5GMN) were obtained from PDB. The hit molecule obtained from virtual screening was docked with the compounds isolated from plant crude methanolic plant extracts of *T. vulgaris* through GC-MS analysis. All structures of the compounds were retrieved from PubChem (https://pubchem.ncbi.nlm.nih.gov/, accessed on 5 July 2022) in SDF format, of which two phytochemicals, i.e., thymol (PubChem: 6989) and carvacrol (PubChem: 10364), showed the highest binding energies with COX-1 and COX-2. The results were presented in energies, and the compounds deemed to be the best candidates for anti-inflammatory activity were those with the lowest docking energies [54].

#### 4.9.2. Swiss ADME Analysis

A free online tool called Swiss ADME was used to determine whether a substance had the characteristics of water-soluble, blood–brain permeable, hepatotoxic, human intestine absorbable, physiochemical, pharmacokinetic, drug-like, and medicinal chemistry. Swiss ADME was used to characterize the top compounds discovered as a consequence of docking results that had the highest inhibitory affinity against COX1 and COX2. It established the toxicity of the substances and characteristics.

#### 4.9.3. Hepatotoxicity

Hepatotoxicity was performed as the protocols given by El-Newary et al. [55] with modifications. Thirty albino rats were taken and divided into five groups for each fraction. Each group contains three rats, such as group 1: control group (normal saline 1 mL/kg of body weight) and groups 2 and 5: plant fractions of ethyl acetate and *n*-butanol with different concentrations of doses (200, 400, 600, and 800 mL/kg body weight) [55,56].

#### 4.9.4. Acute Toxicity

After being treated with fractions, the rats were kept under observation for 24 h to see any signs of acute toxicity. After 24 h, no changes were observed in general behavior, physiological activity, or any deaths. During the experiment, for the biochemical analysis, blood samples were collected after sacrificing the animals, and histopathology was performed on the blood samples. Further, bilirubin, alanine aminotransferase (ALT), the total protein, aspartate aminotransferase (AST), and albumin tests were determined to analyze the biochemical parameters [55].

### 4.10. Effect of HPLC Identified Phytoconstituents on Gastric Cancer Genes

#### 4.10.1. Retrieval of Gastric Cancer Genes

The Human Gene Database (GeneCards, https://www.genecards.org/, accessed on 10 July 2022 version 4.9.0) was used, and gastric cancer genes were identified that cause gastric cancer. This database provides user-friendly and comprehensive knowledge of all annotated and predicted human genes in an integrated and searchable database. Twenty-three gastric genes were retrieved from the Protein Data Bank and visualized by using Discovery Studio version 21.1.0.20298, Dassault systems, Vélizy-Villacoublay, France, accessed on 12 July 2022.

#### 4.10.2. Phytochemical Structure Retrieval (PubChem)

PubChem is an accessible chemical resource of the National Institutes of Health (NIH). It provides data of the chemical structures, physical and chemical characteristics, toxicity, health, and biological activities, as well as the patentability and safety, of all small molecules, as well as macromolecules that have undergone chemical modifications.

#### 4.10.3. Analysis of Venn Diagrams (Bioinformatics and Evolutionary Genomics System)

Bioinformatics and Evolutionary Genomics systems are used to generate targets of drugs and diseases that intersect through a Venn diagram (bioinformatics.psb.ugent.be/webtools/Venn, accessed on 15 July 2022) that shows disease and drug targets intersecting [57].

#### 4.10.4. STRING: Protein-Protein Network Construction

In STRING, 23 target genes were imported from the Venn diagram intersection to construct a PPI network systematically to understand the protein interactions (https://string-db.org/, version 11.0, accessed on 16 July 2022). To construct the PPI network, homo-sapiens were selected, a medium confidence interaction score of 0.40 was set, and all disconnected protein nodes were excluded [57].

#### 4.10.5. Analysis of Molecular Docking

The molecular docking was performed by obtaining the 3D structures of gastric genes from PDB and ligands from PubChem and the protocol as described in the above section.

#### 4.10.6. Molecular Dynamic Simulation

It is a computer-based simulation approach used to analyze the physical motions of atoms or molecules. The molecular dynamics (MD) simulation of hydrogen bond interactions can identify a few significant interactions. Protein docking and virtual screening are made possible by MD simulations. For this work, molecular dynamics simulations were performed using the iMODS server. This server provides access to information about homology-related or macromolecule-related routes that can be explored by normal mode analyses.

#### 4.10.7. Cloud 3D-QSAR Modelling (3-D QSAR)

The Cloud 3D-QSAR Server http://chemyang.ccnu.edu.cn/ccb/server/cloud3dQSAR/ (accessed on 1 August 2022) runs 3D-QSAR jobs by submitting molecular structures and IC_50_ values on the webpage. Molecules are converted into 3D structures, and then, energy minimization is performed. All of them are randomly divided into training sets and test sets, and 3D-QSAR modeling is performed. Force field files are automatically generated, and other results are also automatically analyzed and sorted.

#### 4.10.8. Analyses of Pharmacokinetics and ADME

As a result of docking, SwissADME characterized the compound with the highest binding affinity against gastric cancer genes. By using computer-aided tools, it determined the lipophilicity, pharmacokinetics, and drug-likeness of the drug, as well as its absorption, distribution, solubility, and toxicity.

### 4.11. Cytotoxicity Analysis

A major bioactive compound that exhibits high binding affinity, chlorogenic acid, was submitted in SMILES format to the cell line cytotoxicity predictor (CLC-Pred) (http://www.way2drug.com/Cell-line/, accessed on 5 August 2022) to predict its toxicity. The probability of activity (Pa) and probable inactivity (Pi) were used to represent the output predicted activity.

### 4.12. Hemolytic Activity

A 5-mL human blood sample was obtained from healthy volunteers in EDTA vials. The blood was centrifuged, the supernatant was discarded, and the pellet was washed with 150 mM NaCl. The erythrocyte suspension was made in sterile phosphate buffer (PBS) chilled at 4 °C by making up to 20 mL volume of blood suspension. In microcentrifuge tubes, 0.8 mL of plant fractions were combined with 0.2 mL of blood suspension at different 12–0.37 mg/mL concentrations. A 30-min incubation was performed at 37 °C, followed by 15-minute centrifugation at 16,000 rpm, and then, 100 μL of the supernatant was collected and mixed with 900 μL PBS to measure the absorbance in the ELISA reader at 630 nm. PBS was used as the negative control, and 0.1% Triton X-100 was used as a positive control [58].
**% Hemolysis** = Abs of control−Abs of sample/Abs of control × 100

### 4.13. Thrombolytic Activity

Different fractions were assessed for their thrombolytic activity by the method reported by Ramjan et al. [59], and 500 μL of fresh human blood was added to already measured empty microcentrifuge tubes and incubated for 60 min at 37 °C. The serum was removed after incubation, and the weight was recalculated. It showed the weight of the clot. Streptokinase SK was used as the positive control, and deionized water was added as the negative control, and 100 μL of extracts and SK suspension were used for the in vitro analysis and added in incubated pre-weighted tubes and then again incubated for a further 90 min. After incubation, the serum was removed, weighed, and the lysis was calculated by the formula, with 100 μL of distilled water used as the negative control [59]:**%Lysis** = **W_C_/W_L_** × **100**
**W_C_** = weight of lysis and **W_L_** = weight of clot

### 4.14. Antipyretic Activity In Vivo

Antipyretic activity was performed on albino rats to induce fever by giving yeast in a normal saline injection. In this, rats were divided as per discussed by anti-inflammatory and brewer yeast in a normal saline injection given to the standard and experimental group rats below the nape of the neck. After an interval of 2 h, pyrexia developed, and a dose of 50, 100, 200, or 400 mg/mL was given to the experimental rats, and the standard were treated with paracetamol. The temperature was noted at 0-h, 30-min, 60-min, and 120-min time intervals. The antipyretic activity was calculated by the given formula [59]:**% Reduction** = T − Cb/T − A × 100
where T = temperature after pyrexia, Cb = temperature of experimental groups 2–7, and A = normal body temperature.

### 4.15. Statistical Results

The results are expressed using the standard mean deviation (SEM). One-way ANOVA was used to compare the differences between the treated groups and control group, and then Dunnett’s post hoc test was performed. SPSS was used to conduct the analysis, and *p* > 0.05 was regarded as significant.

## 5. Conclusions

The aforesaid study showed the in vitro and in vivo antioxidant, anti-bacterial, hemolytic, thrombolytic, anti-inflammatory, and anti-diabetic activity of *Thymus vulgaris* ethyl acetate and *n*-butanol fractions. The GC-MS analysis of methanolic extract provided identification of thymol, carvacrol, p-cymene and other potential phytoconstituents which showed strong biological activities. The total phenolic and flavonoid content was determined, and results showed that ethyl acetate contains high TPC and TFC. HPLC analysis revealed that ethyl acetate contains a high amount of phenolics and flavonoids, showing strong antioxidant, anti-bacterial and anti-inflammatory activities. Fourier transform infrared spectroscopy analysis shows the presence of a functional group that confirms the phytoconstituents identified through GC-MS and HPLC. In biological activities, ethyl acetate shows strong antioxidant and anti-bacterial activities as phenolic, and flavonoids attributed many of the antioxidant properties to the compounds due to their strong hydrogen donating ability. High binding energy was obtained for thymol, carvacrol, p-Cymene and eugenol, which revealed that these compounds synergistically inhibit COX1, COX2 and prevent inflammation. Hepatotoxicity results revealed that ethyl acetate and *n*-butanol fractions show no toxicity and high dose and safe for oral administrations. Phytoconstituents of these fractions could be candidates for drug development. The phenolic compounds obtained through HPLC analysis as a ligand were performed against gastric cancer genes in silico and the results revealed that chlorogenic acid is a strong candidate for drug development as it shows high binding energy and good ADMET properties. Therefore, these findings suggest that *T. vulgaris* is an excellent candidate for the treatment of inflammation and different human relative ailment. The pharmacological and medicinal potential of *T. vulgaris* showed that it is quite auspicious as a versatile therapeutic plant, and it should be further investigated.

## Figures and Tables

**Figure 1 molecules-27-08512-f001:**
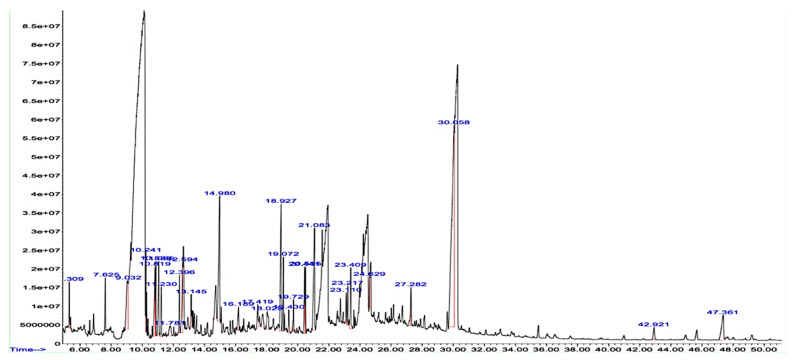
GC-MS analysis of a methanolic extract of *T. vulgaris*.

**Figure 2 molecules-27-08512-f002:**
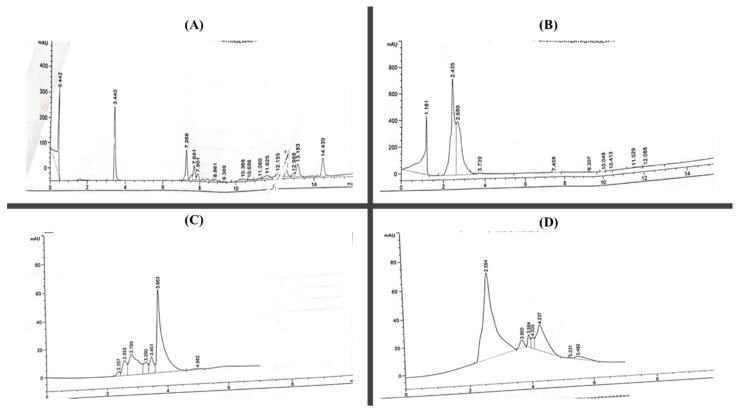
Chromatogram of phenolics and flavonoids of fractions of *T. vulgaris.* (**A**) Phenolics of ethyl acetate, (**B**) phenolic of *n*-butanol, (**C**) flavonoids of ethyl acetate, and (**D**) flavonoids of *n*-butanol.

**Figure 3 molecules-27-08512-f003:**
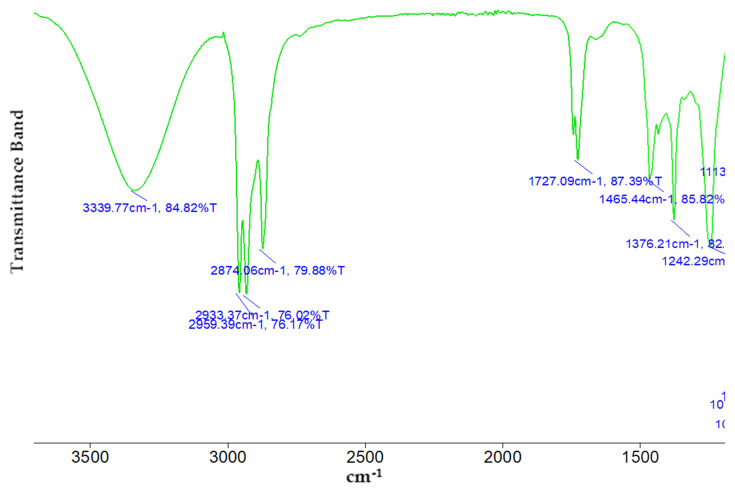
FTIR spectrum analysis of the methanolic extract of the *T. vulgaris* plant.

**Figure 4 molecules-27-08512-f004:**
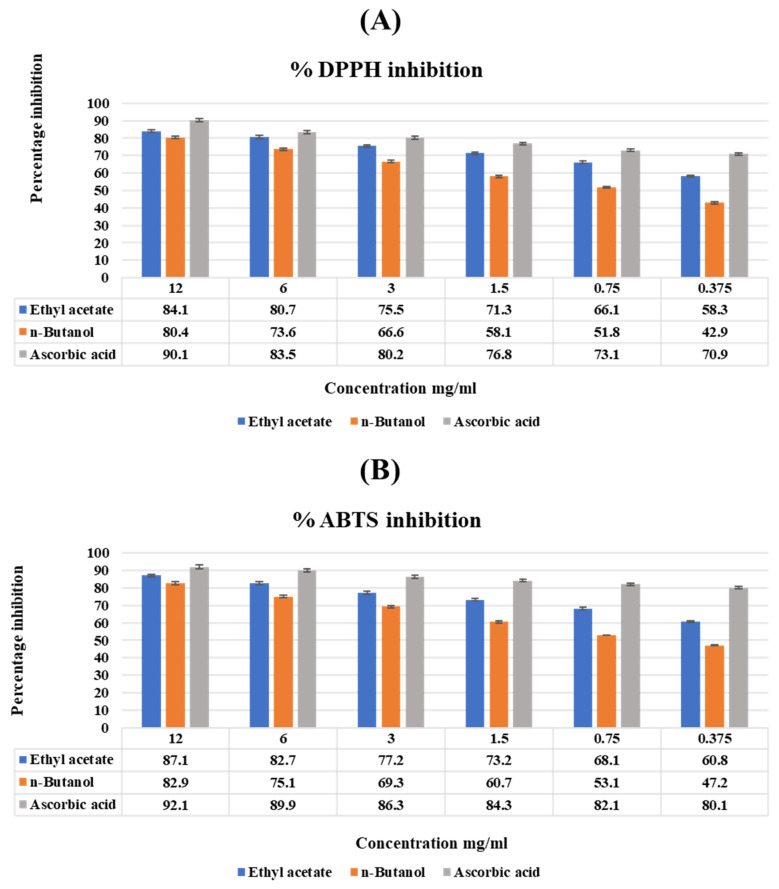
(**A**) Antioxidant activity of fractions of *T. vulgaris* through the DPPH assay. (**B**) Antioxidant activity of fractions of *T. vulgaris* through the ABTS assay. Vertical bars represent the mean ± S.E. The values are expressed in the mean ± SD of triplicates, and all values were statistically significant at *p* < 0.05.

**Figure 5 molecules-27-08512-f005:**
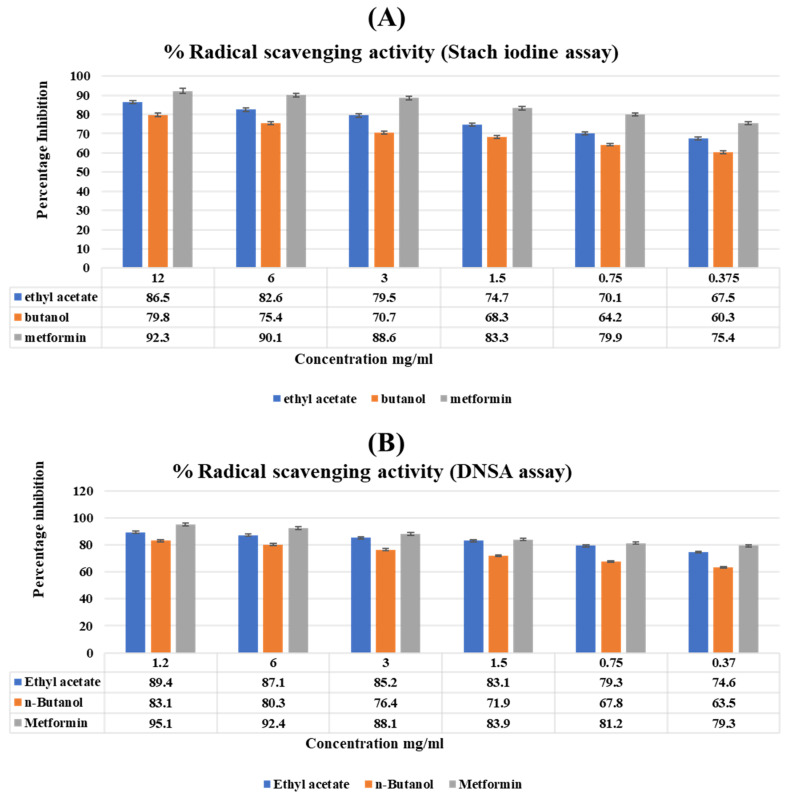
Showing the antidiabetic activity of the fractions of *T. vulgaris* plant extract. Vertical bars represent the mean ± S.E. The values were expressed in the mean ± SD of triplicates, and all values were statistically significant at *p* < 0.05. (**A**) Starch iodine assay. (**B**) DNSA assay.

**Figure 6 molecules-27-08512-f006:**
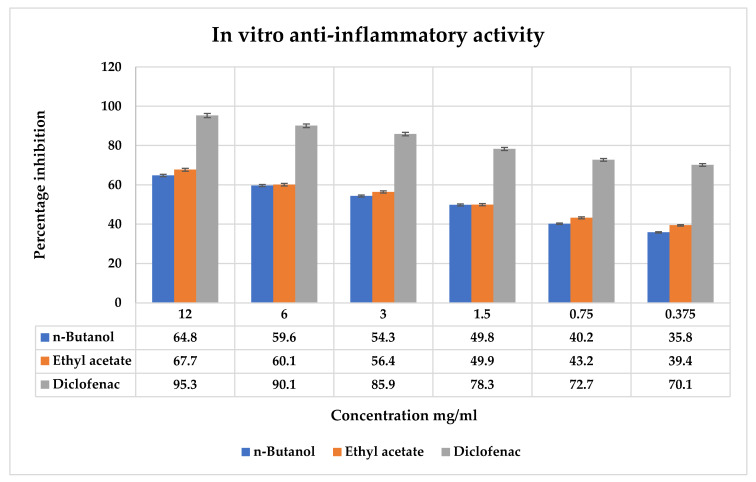
Anti-inflammatory activity of ethyl acetate and *n*-butanol fractions of *T. vulgaris* compared with the standard diclofenac. For each parameter, the vertical bar represents the mean ± S.E. The values were expressed in the mean ± SD of triplicates, and all values were statistically significant at *p* < 0.05.

**Figure 7 molecules-27-08512-f007:**
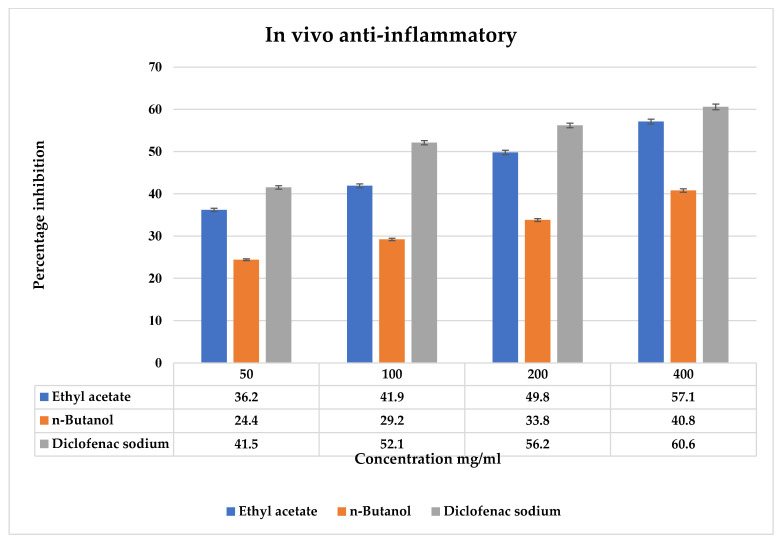
In vivo anti-inflammatory activity of the fractions of *T. vulgaris*. The vertical bar represents the mean ± S.E. The values were expressed in the mean ± SD of triplicates, and all values were statistically significant at *p* < 0.05.

**Figure 8 molecules-27-08512-f008:**
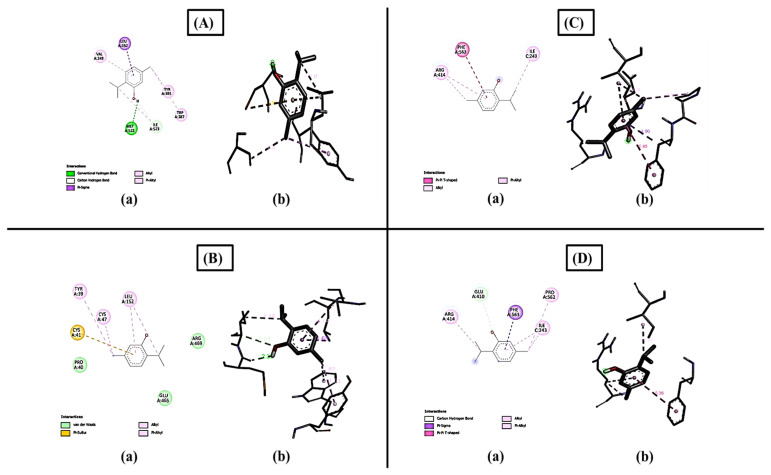
Binding residues of compounds with COX1 and COX2: (**A**) represents the thymol compound docked with COX1 (−6.4), (**a**) shows the 2D image of the compound and (**b**) shows the 3D image of the compound with binding residues, (**B**) represents docked complex of carvacrol with COX1(−6.3): (**a**) shows the 2D image of the docked compound and (**b**) shows the 3D image of the docked compound with binding amino acid residues, (**C**) represents the docked complex of thymol with COX2 (−6.3): (**a**) shows the 2D diagram of the docked compound and (**b**) shows the 3D image of the docked compound with binding amino acids, and (**D**) represents the docked complex of carvacrol with COX2 (−6.2): (**a**) shows the 2D image of the carvacrol docked compound with COX2 and (**b**) shows the 3D image of the docked compound with binding amino acids.

**Figure 9 molecules-27-08512-f009:**
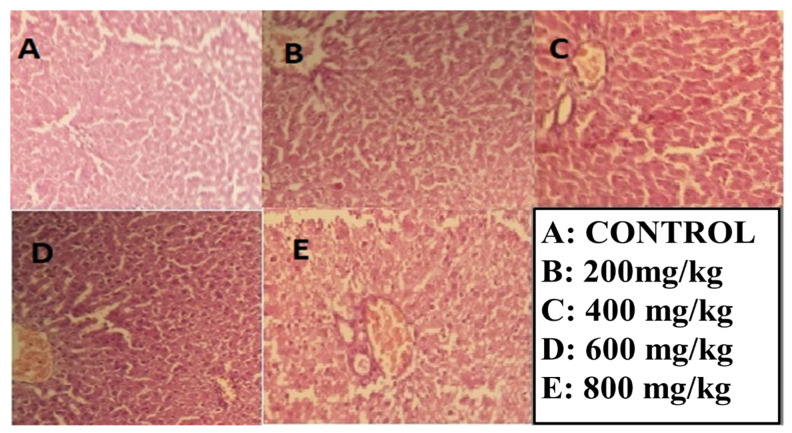
Hepatotoxicity histopathology investigations of the liver section given the ethyl acetate fraction (**A**) No fibrosis with no accumulation of immune cells (**B**) Slight fibrosis with no ballooning (**C**) nuclear degeneration, and no accumulation of immune cells (**D**) Slight portal activity, slight infiltration of macrophages nuclear variation (**E**) no vacuolar degeneration, and hydropic degeneration present.

**Figure 10 molecules-27-08512-f010:**
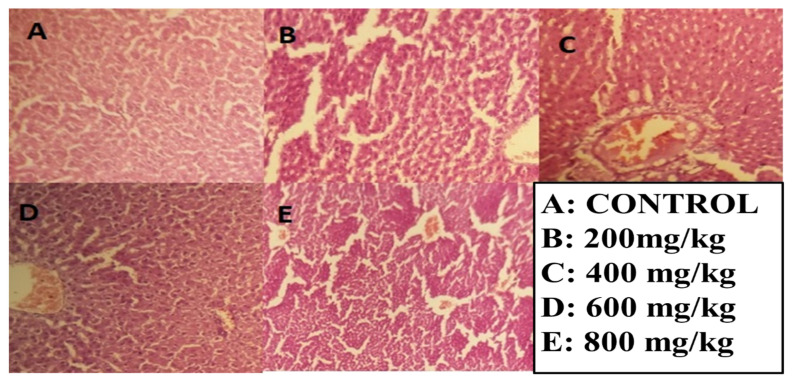
Hepatotoxicity histopathology investigations of the liver section given the *n*-butanol fraction (**A**) No ballooning of cells and no fibrosis (**B**) Slightly ballooning of hepatocytes and no accumulation of ECM (**C**) Overall, the liver tissue seems healthy (**D**) Slightly ballooning, accumulation of ECM was observed, nuclear variation (**E**) no vacuolar degeneration, and fatty droplets are present.

**Figure 11 molecules-27-08512-f011:**
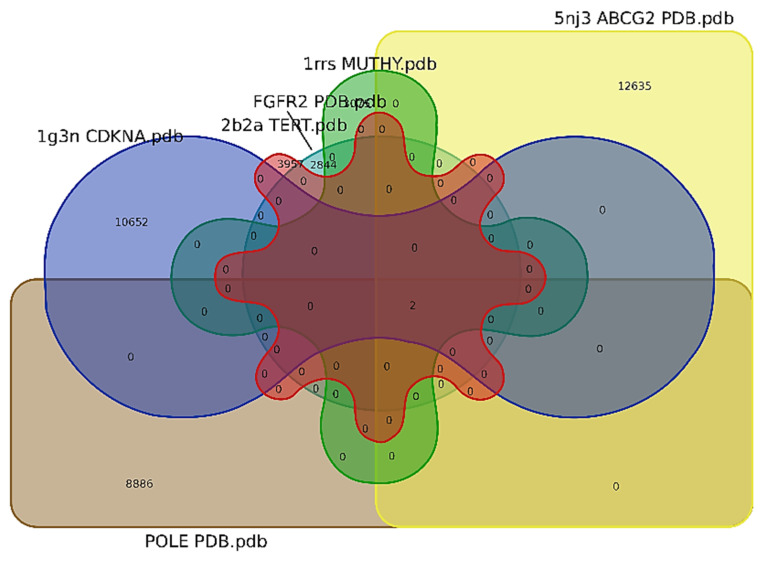
A Venn diagram showing the intersection of identified compounds with several elements: ABCG2 (12,635), CDKNA (10,652), POLE (8886), FGFR2 (2844), TRET (3957), and MUTHY (3075).

**Figure 12 molecules-27-08512-f012:**
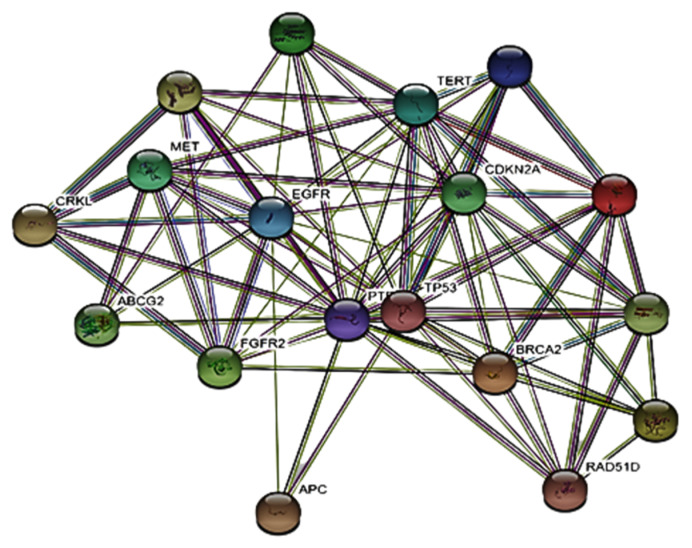
The PPI network of gastric cancer genes. Each node represents the relevant gene, and the edges represent protein–protein associations.

**Figure 13 molecules-27-08512-f013:**
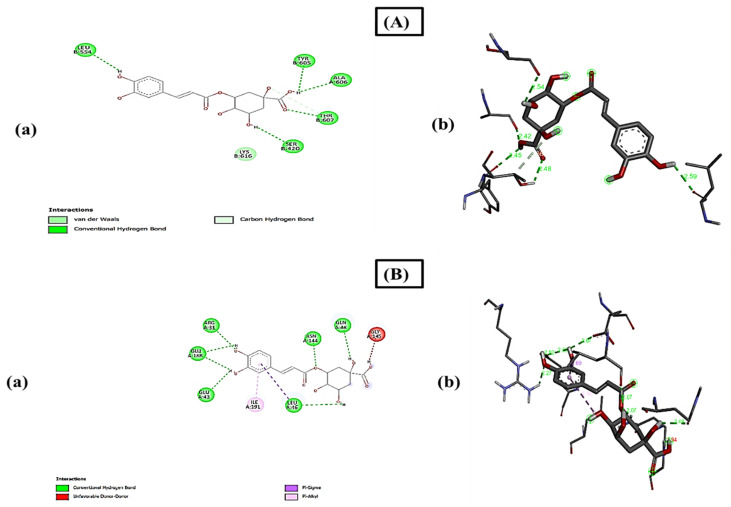
Molecular interactions. (**A**) Chlorogenic acid and ABCG2 (−8). (**a**) Two-dimensional image of docked complex. (**b**) Three-dimensional image of docked complex. (**B**) Chlorogenic acid and Muthy (−8.8). (**a**) Two-dimensional image of docked complex. (**b**) Three-dimensional image of docked complex.

**Figure 14 molecules-27-08512-f014:**
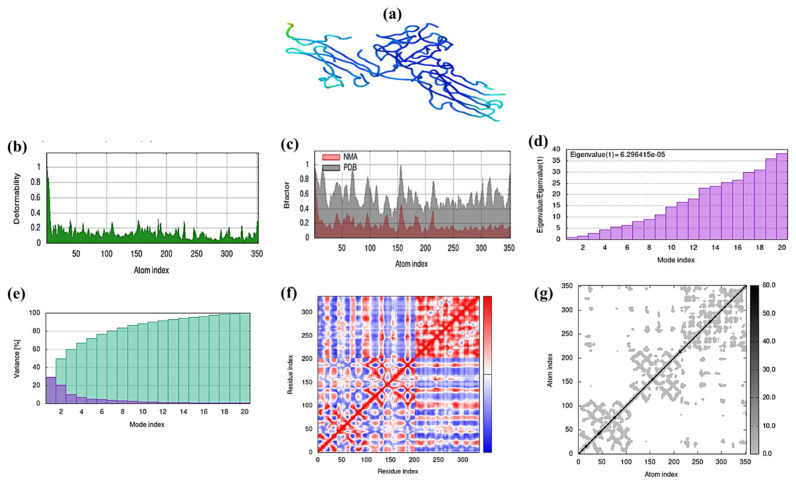
Molecular Dynamics Simulations of the Chlorogenic acid docked with gene Muthy (**a**) shows the docked compound (**b**) shows the deformability, which indicates a low level of deformation at all the residues (**c**) shows the B-factor, (**d**) Eigon values are shown, and (**e**) indicates the variance explained in both purple and green (**f**) and (**g**) show the covariance and elastic network of the complex.

**Figure 15 molecules-27-08512-f015:**
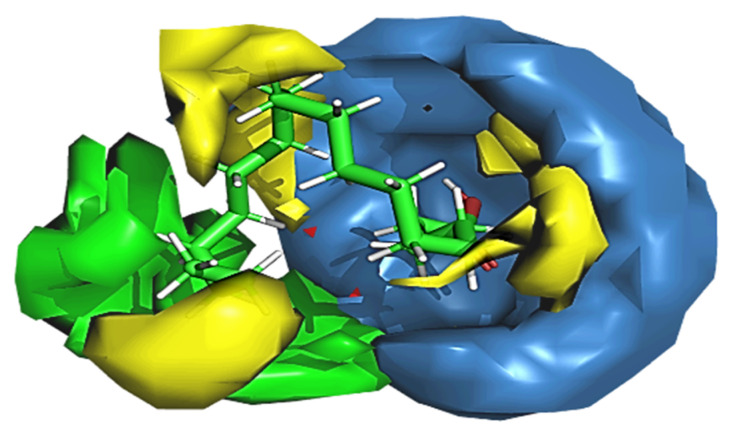
Contour map of best hit compound chlorogenic acid.

**Figure 16 molecules-27-08512-f016:**
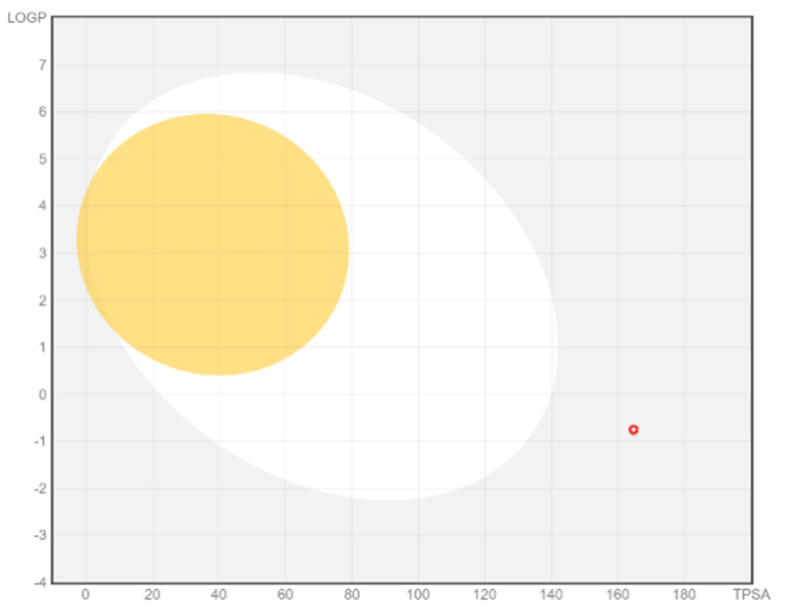
BOILED-Egg image of chlorogenic acid. The yellow region shows a high probability of brain penetration, the white region shows high permeability of passive absorption by the gastrointestinal tract and the red dot shows that compound that is not the substrate of P-gp.

**Figure 17 molecules-27-08512-f017:**
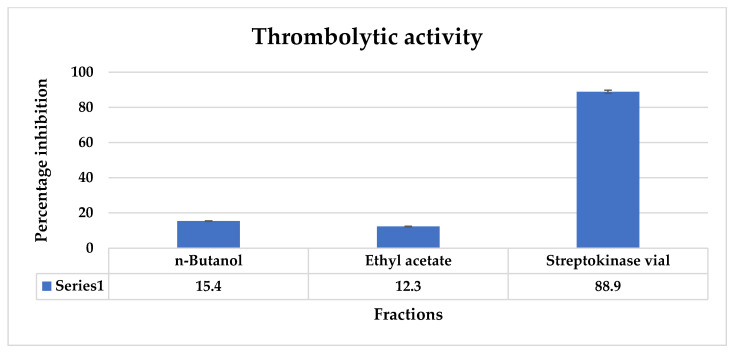
The thrombolytic activity of ethyl acetate and *n*-butanol fraction of *T. vulgaris*. Values are expressed in mean ± SD and all values are significant at *p* < 0.05.

**Figure 18 molecules-27-08512-f018:**
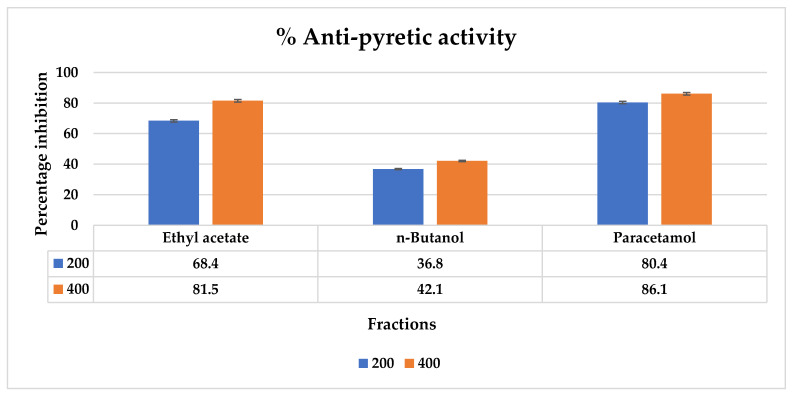
The antipyretic activity of fractions of *T. vulgaris* fractions in vivo.

**Table 1 molecules-27-08512-t001:** GC-MS analysis of a methanolic extract of *T. vulgaris*.

Sr. No	Compound	Molecular Weight	% Area	Compound Nature	Retention Index
1	Mequinol	124.14	1.21	Methoxyphenol	1183
2	Isothymol-methyl ether	164.24	1.56	Alkylbenzene	1244
3	Eugenol	165.20	4.66	Phenylpropanoid	1327.7
4	2-Methoxy-4-vinylphenol	150.17	1.17	Phenol	1315
5	2-Allyl-4-methylphenol	148.20	3.47	Phenol	NA
6	9-Octadecyne	250.5	2.16		NA
7	3-Methyl-4-isopropylphenol	150.22	2.95	Phenol	1334
8	(-)-Spathulenol	220.351	2.03	Tricyclic sesquiterpenes alcohol	1576
9	Vanillin	152.15	1.09	Phenolic aldehyde	NA
10	Aromandendrene	204.35	1.82	Sesquiterpenoid	1455
11	p-Cymene-	166.22	6.68	Terpenoid	1021
12	Phenol,2,6-dimethoxy-	154.16	1.03	Phenol	1347
13	Carvacrol	150.22	7.77	Terpenoid	1274
14	Benzenepropanol, 4-hydroxy-3-methoxy-	182.22	1.13	Phenol	1660.6
15	7-Hydroxyfarnesen	220.35	1.16		2273
16	Tetradecanoic acid	228.4	1.51	Saturated fatty acid	1763
17	Linoleic acid methyl ester	294.5	6.01	Polyunsaturated omega-6 fatty acid	2066.9
18	Neophytadiene	278.524	2.50	Alkene	NA
19	Heptadecanal	254.5	0.90	Long-chain fatty aldehyde	1897
20	Bis(2-ethylhexyl) phthalate	390.6	1.00	Plasticizer	2492.6
21	Adamantane	136.23	2.58	Polycyclic alkane.	1121
22	Hexadecanoic acid, methyl ester	270.4507	1.86	Fatty acid methyl ester	1904.1
23	Phthalic acid	320.4232	3.40	Benzenedicarboxylic acid	1643
24	Isopropyl myristate	270.5	1.02	Isopropyl alcohol	1812.8
25	linolenelaidic acid methyl ester	292.5	1.39	Omega-3-fatty acids	NA
26	Phytol	296.5	2.55	Diterpenoid	2122
27	Stearic acid	284.5	1.02	Saturated fatty acid	2179
28	Borneol	154.25	1.52	Terpene derivative	NA
29	Thymol	150.22	28.88	Terpenoid	1265.51
30	Vitamin E	430.7	0.88	Fat-soluble vitamin	NA
31	Gamma-sitosterol	414.7067	3.06	Phytosterols	NA

**Table 2 molecules-27-08512-t002:** HPLC analysis of phenolics and flavonoids of *T. vulgaris* ethyl acetate and *n*-butanol fractions.

Fractions	Phytochemicals	Concentration mg/g	Retention Time
**Phenolics**			
Ethyl acetate	Chlorogenic acid	0.012	9.36
	Caffeic acid	0.151	10.36
	Sinapic acid	0.115	11.62
	Benzoic acid	0.343	12.98
	Vanillic acid	0.144	13.18
*n*-Butanol	Chlorogenic acid	0.044	9.307
	Caffeic acid	0.154	10.41
	Sinapic acid	0.046	11.52
Flavonoids			
Ethyl acetate	Myricetin	0.204	2.78
	Quercetin	0.811	3.63
*n*-Butanol	Quercetin	0.473	3.66
	Kaempferol	0.109	4.23

**Table 3 molecules-27-08512-t003:** FTIR interpretation of compounds of methanolic extract of *Thymus vulgaris*.

Sr. No	Wave Number cm^−1^	Functional Groups	Identified Phytocompounds
1	3339	O-H stretch	Alcohols, phenols
2	2933	-CH stretch	Saturated aliphatic compounds
3	2959	O-H stretch, carboxylic group	Carboxylic acid
4	2874	O-H stretch, carboxylic group	Carboxylic acid
5	1465	C=C-C, Aromatic ring	Aromatic compounds
6	1376	Alcoholic group, O-H bend	Tertiary alcohols or phenols
7	1242	C-O stretch	Acids

**Table 4 molecules-27-08512-t004:** Effects of treatment with chloroform and ethyl acetate fractions of *T. vulgaris* on the liver function test parameters.

Sr. No	Fractions	Liver Function Parameters	Control	200 mg/kg	400 mg/kg	600 mg/kg	800 mg/kg
1	Ethyl acetate	ALP (U/L)	166.31 ± 1.94	165.7 ± 1.8	162 ± 1.7	157.9 ± 1.61	143.74 ± 1.32
		AST (U/L)	66.42 ± 0.7	64.6 ± 0.66	61.3 ± 0.6	57.58 ± 0.55	53.83 ± 0.5
		ALT (U/L)	36.55 ± 0.42	34.7 ± 0.39	30.95 ± 0.3	29.59 ± 0.25	27.48 ± 0.27
		Total protein	8.1 ± 0.04	8.6 ± 0.08	8.9 ± 0.05	9.5 ± 0.1	9.8 ± 0.13
		Globulin	3.9 ± 0.09	4.1 ± 0.07	4.6 ± 0.05	5.1 ± 0.1	5.8 ± 0.07
		Albumin	4.75 ± 0.05	4.77 ± 0.04	4.68 ± 0.11	4.51 ± 0.07	4.4 ± 0.02
2	*n*-Butanol	ALP (U/L)	155 ± 2.3	153 ± 0.2	146 ± 1.4	141 ± 1.4	139.34 ± 1.42
		AST (U/L)	63.83 ± 0.65	60.4 ± 0.6	56.2 ± 0.52	52.6 ± 0.48	49.1 ± 0.55
		ALT (U/L)	31.63 ± 0.4	29.4 ± 0.32	24.82 ± 0.21	22.1 ± 0.2	21.4 ± 0.18
		Total protein	7.9 ± 0.07	8.1 ± 0.05	8.4 ± 0.12	8.9 ± 0.19	9.13 ± 0.11
		Globulin	4.61 ± 0.1	4.41 ± 0.04	3.95 ± 0.1	3.51 ± 0.11	3.23 ± 0.03
		Albumin	4.95 ± 0.09	4.4 ± 0.08	3.9 ± 0.11	3.3 ± 0.08	2.8 ± 0.02

Values were expressed in mean ± SD of 3 replicates, the results were compared by two-way ANOVA, and all the values were statistically significant at *p* < 0.05.

**Table 5 molecules-27-08512-t005:** CLC-Pred analysis of chlorogenic acid cancerous cell line and non-cancerous cell line prediction.

Sr. No	Pa	Pi	Cell Line	Cell Line-Full Name	Tissue	Tumor Type
1	0.581	0.023	NCI-H838	Non-small cell lung cancer, 3rd stage	Lung	Carcinoma
2	0.525	0.020	HepG2	Hepatoblastoma	Liver	Hepatobloastoma
3	0.459	0.023	MDA-MB-453	Breast adinocarcinoma	Breast	Carcinoma
4	0.436	0.046	YAPC	Pancreatic carcinoma	Pancreas	Carcinoma
5	0.228	0.036	BGC-823	Stomach adenocarcinoma	Stomach	Carcinoma
6	0.243	0.123	MKN-7	Gastric carcinoma	Stomach	Carcinoma
Non-cancerous cell line						
7	0.545	0.018	WI-38 VA13	Embryonic lung fibroblast	Lung	
8	0.282	0.034	HEL299	Fibroblast	Lung	
9	0.156	0.057	IMR-90	Embryonic lung fibroblast	Lung	
10	0.101	0.077	HUVEC	Umbilical vein endothelial cell	Endothelium	
11	0.070	0.066	HaCaT	Keratinocyte	Skin	

**Table 6 molecules-27-08512-t006:** Percentage hemolysis of fractions of *T. vulgaris*.

Sr. No	Concentration mg/mL	Ethyl Acetate	*n*-Butanol	Triton X-100
1	12	9.6	9.6	96.1
2	6	8.3	8.3	
3	3	6.8	6.8	
4	1.5	4.2	4.2	
5	0.75	3.1	3.1	
6	0.37	0.7	1.9	

All samples were taken in triplicates and are significant *p* > 0.05.

## Data Availability

Not applicable.

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
