# Peer review of "HPLC, FTIR and GC-MS Analyses of Thymus vulgaris Phytochemicals Executing In Vitro and In Vivo Biological Activities and Effects on COX-1, COX-2 and Gastric Cancer Genes Computationally"

_molecules, 2022, doi:10.3390/molecules27238512_

Round 1

Reviewer 1 Report

Regarding the MS based on my following concerns I cannot recommend for further consideration:

- Thymus vulgaris is a very well-known plant, whereas extensive biological and phytochemical investigations have been previously done on this plant: lack of novelty

- most of the biological assays have been done on this plant formerly: lack of enough novelty

- it is not properly written, e.g. the title "HPLC and GC-MS Quantification of Phytoconstituents from T. vulgaris eliciting the potential of bioactive compounds by executing multiple in-vitro and in vivo biological activities inducing functionalized capabilities on COX-1, COX-2 and gastric cancer gen", too long; or Abstract

- the most important issue, the study is not well-designed, a bunch of bioassays with no aim, I would recommend targeting one bioassay and study their bioactive phytochemicals

Author Response

Dear Ms.  Karol Zhang

Assistant Editor

Molecules

Please find the attached response to the reviewer comments on our article entitled “ HPLC and GC-MS Quantification of Phytoconstituents from T. vulgaris eliciting the potential of bioactive compounds by executing multiple in-vitro and in vivo biological activities inducing functionalized capabilities on COX-1, COX-2 and gastric cancer genes computationally” with manuscript ID molecules-1995076.

Reviewer 1

Comments and Suggestions for Authors

Regarding the MS based on my following concerns I cannot recommend for further consideration:

- Thymus vulgaris is a very well-known plant, whereas extensive biological and phytochemical investigations have been previously done on this plant: lack of novelty.

AR: Thank You for your comment. Yes, it is a well-known plant and extensive studies have been done on it but following is the work that has not been performed previously

  • The fractions of n-Butanol and Ethyl Acetate of this plant had not been identified previously via GC-MS and HPLC quantification.
  • FTIR analysis of functional groups interpretation of Thymus vulgaris has not yet be done in previous research.
  • In Addition in-vivo trials of this plant are not yet performed.
  • Further the activities like anti-diabetic, anti-hemolytic, anti-pyretic, thrombolytic and hepatotoxicity analysis had not yet being interpreted, which are well described in this manuscript proving the enhanced potential of Thymus vulgaris.
  • Lastly, the computational analysis on inflammatory (COX-1 and COX-2) and gastric genes is not evaluated yet.
  • Moreover, chlorogenic acid is the compound having enhanced potential against gastric cancer, analyzed via HPLC.
  • The antioxidant analysis is being performed for the essential oil of Thymus vulgaris and not for simple plant extract of this plant.
  • Furthermore, chlorogenic acid is the compound analyzed by HPLC analysis that has a neutralizing effect against gastric cancer, interpreted computationally via docking analysis of the particular compounds with gastric cancer-causing genes.

- most of the biological assays have been done on this plant formerly: lack of enough novelty.

AR: Thank You for your comment. Yes, I agree the biological assays like antioxidant, anti-bacterial and anti-inflammatory has been done, but in-vivo trials of these activities has not been performed yet. Though these trials have been done in the current manuscript. Moreover, anti-diabetic, anti-hemolytic, anti-pyretic, thrombolytic and hepatotoxicity analysis of Thymus vulgaris are unidentified yet, which are well described in this manuscript along with its computational interpretation.

- it is not properly written, e.g. the title "HPLC and GC-MS Quantification of Phytoconstituents from T. vulgaris eliciting the potential of bioactive compounds by executing multiple in-vitro and in vivo biological activities inducing functionalized capabilities on COX-1, COX-2 and gastric cancer gen", too long; or Abstract.

AR: Thank You for your comment. The title and abstract of the manuscript have been shortened. [See title and abstract in the revised manuscript highlighted in RED].

- the most important issue, the study is not well-designed, a bunch of bioassays with no aim, I would recommend targeting one bioassay and study their bioactive phytochemicals.

AR: Thank You for your comment. In the current manuscript the targeted bioactive compounds are of Thymus vulgaris plant, of which potency has been analyzed by performing various bioassays. We really appreciate your recommendation regarding targeting one bioassay but the reason of performing multiple bioassays is that we have analyze the potential and toxicity in every aspect i.e., antioxidant, anti-diabetic, anti-inflammatory, anti-hemolytic, anti-pyretic, thrombolytic and hepatotoxicity analysis. These bioassays showed that Thymus vulgaris has potential against these all-biological assays.

Regards

Dr. Tariq Aziz (Postdoc, PhD)

Associate Professor

Reviewer 2 Report

The work presented is good enough. need some grammar error to be removed. idea is well designed . will rise impact of journal as well as will give new way to researchers.

Author Response

Dear Ms.  Karol Zhang

Assistant Editor

Molecules

Please find the attached response to the reviewer comments on our article entitled “ HPLC and GC-MS Quantification of Phytoconstituents from T. vulgaris eliciting the potential of bioactive compounds by executing multiple in-vitro and in vivo biological activities inducing functionalized capabilities on COX-1, COX-2 and gastric cancer genes computationally” with manuscript ID molecules-1995076.

Reviewer 2

Comments and Suggestions for Authors

The work presented is good enough. need some grammar error to be removed. idea is well designed . will rise impact of journal as well as will give new way to researchers.

AR: Thank you very much for your comments. All the grammatical mistakes have been removed and the English language of the manuscript has been rechecked by a native speaker. We really appreciate your comments. Thanks a million once again.

Regards

Dr. Tariq Aziz (Postdoc, PhD)

Associate Professor

Reviewer 3 Report

I am surprised that the authors submitted this manuscript to Molecules while knowing all they are reporting is already known. Starting from the title, it does not make sense: what does it mean by "executing multiple in-vitro and in vivo biological activities..."? You do not execute but rather evaluate activities! I read the title several times but still do not know what to expect in the manuscript. I still read the manuscript and could not help but notice all the authors presented in this manuscript is already known. Even if a slight bit of new info is described, it still does not warrant a manuscript in Molecules.

Below, I will cite the papers on various topics that have already been reported in much better detail than the authors said here:

GC-MS Analysis:

- GC/MS evaluation of thyme (Thymus vulgaris L.) oil composition and variations during the vegetative cycle. Journal of Pharmaceutical and Biomedical Analysis, 2002, 29(4), 691-700. https://doi.org/10.1016/S0731-7085(02)00119-X

- Gas chromatography-mass spectrometry analysis of different organic crude extracts from the local medicinal plant of Thymus vulgaris L. Asian Pac J Trop Biomed. 2013, 3(1):69-73. doi: 10.1016/S2221-1691(13)60026-X. 

- GC-MS analysis and biological activities of Thymus vulgaris and Mentha arvensis essential oil. Turkish Journal of Biochemistry, 2019, 44(3). https://doi.org/10.1515/tjb-2018-0258.

HPLC Analysis:

- A validated high performance liquid chromatography method for the analysis of thymol and carvacrol in Thymus vulgaris L. volatile oil. Pharmacogn Mag. 2010, 6(23):154-8. doi: 10.4103/0973-1296.66927.

- Polyphenolic and molecular variation in Thymus species using HPLC and SRAP analyses. Sci Rep, 2021, 11, 5019. https://doi.org/10.1038/s41598-021-84449-6.

Bioactivities:

- Effects of Thyme Extract Oils (from Thymus vulgaris, Thymus zygis, and Thymus hyemalis) on Cytokine Production and Gene Expression of oxLDL-Stimulated THP-1-Macrophages. J Obes. 2012;2012:104706. doi: 10.1155/2012/104706.

- Thymus vulgaris Essential Oil and Its Biological Activity. Plants. 2021, 19, 10(9), 1959. doi: 10.3390/plants10091959.

- The Potential Gastrointestinal Health Benefits of Thymus Vulgaris Essential Oil: A Review. Biomed Pharmacol J. 2019, 12(4). https://dx.doi.org/10.13005/bpj/1810

Computational Studies

-Flurbiprofen–antioxidant mutual prodrugs as safer nonsteroidal anti-inflammatory drugs: Synthesis, pharmacological investigation, and computational molecular modeling. Drug Des Devel Ther. 2016;10:2401-2419. https://doi.org/10.2147/DDDT.S109318.

- Natural Dietary Supplement, Carvacrol, Alleviates LPS-Induced Oxidative Stress, Neurodegeneration, and Depressive-Like Behaviors via the Nrf2/HO-1 Pathway. J Inflamm Res. 2021;14:1313-1329. https://doi.org/10.2147/JIR.S294413.

Author Response

Dear Ms.  Karol Zhang

Assistant Editor

Molecules

Please find the attached response to the reviewer comments on our article entitled “ HPLC and GC-MS Quantification of Phytoconstituents from T. vulgaris eliciting the potential of bioactive compounds by executing multiple in-vitro and in vivo biological activities inducing functionalized capabilities on COX-1, COX-2 and gastric cancer genes computationally” with manuscript ID molecules-1995076.

                                                Reviewer 3

Comments and Suggestions for Authors

I am surprised that the authors submitted this manuscript to Molecules while knowing all they are reporting is already known. Starting from the title, it does not make sense: what does it mean by "executing multiple in-vitro and in vivo biological activities..."? You do not execute but rather evaluate activities! I read the title several times but still do not know what to expect in the manuscript. I still read the manuscript and could not help but notice all the authors presented in this manuscript is already known. Even if a slight bit of new info is described, it still does not warrant a manuscript in Molecules.

Below, I will cite the papers on various topics that have already been reported in much better detail than the authors said here:

GC-MS Analysis:

- GC/MS evaluation of thyme (Thymus vulgaris L.) oil composition and variations during the vegetative cycle. Journal of Pharmaceutical and Biomedical Analysis, 2002, 29(4), 691-700. https://doi.org/10.1016/S0731-7085(02)00119-X

- Gas chromatography-mass spectrometry analysis of different organic crude extracts from the local medicinal plant of Thymus vulgaris L. Asian Pac J Trop Biomed. 2013, 3(1):69-73. doi: 10.1016/S2221-1691(13)60026-X. 

- GC-MS analysis and biological activities of Thymus vulgaris and Mentha arvensis essential oil. Turkish Journal of Biochemistry, 2019, 44(3). https://doi.org/10.1515/tjb-2018-0258.

HPLC Analysis:

- A validated high performance liquid chromatography method for the analysis of thymol and carvacrol in Thymus vulgaris L. volatile oil. Pharmacogn Mag. 2010, 6(23):154-8. doi: 10.4103/0973-1296.66927.

- Polyphenolic and molecular variation in Thymus species using HPLC and SRAP analyses. Sci Rep, 2021, 11, 5019. https://doi.org/10.1038/s41598-021-84449-6.

Bioactivities:

- Effects of Thyme Extract Oils (from Thymus vulgaris, Thymus zygis, and Thymus hyemalis) on Cytokine Production and Gene Expression of oxLDL-Stimulated THP-1-Macrophages. J Obes. 2012;2012:104706. doi: 10.1155/2012/104706.

- Thymus vulgaris Essential Oil and Its Biological Activity. Plants. 2021, 19, 10(9), 1959. doi: 10.3390/plants10091959.

- The Potential Gastrointestinal Health Benefits of Thymus Vulgaris Essential Oil: A Review. Biomed Pharmacol J. 2019, 12(4). https://dx.doi.org/10.13005/bpj/1810

Computational Studies

-Flurbiprofen–antioxidant mutual prodrugs as safer nonsteroidal anti-inflammatory drugs: Synthesis, pharmacological investigation, and computational molecular modeling. Drug Des Devel Ther. 2016;10:2401-2419. https://doi.org/10.2147/DDDT.S109318.

- Natural Dietary Supplement, Carvacrol, Alleviates LPS-Induced Oxidative Stress, Neurodegeneration, and Depressive-Like Behaviors via the Nrf2/HO-1 Pathway. J Inflamm Res. 2021;14:1313-1329. https://doi.org/10.2147/JIR.S294413.

AR: Thank you very much for your comment. We really appreciate your point of view and do consider the above mentioned literature. In comparison to above literature, there  are few points we want to mention which makes our manuscript findings different and novel:

GC-MS AND HPLC ANALYSIS

  • The fractions of n-Butanol and Ethyl Acetate of this plant had not been identified previously via GC-MS and HPLC quantification.
  • FTIR analysis of functional groups interpretation of Thymus vulgaris has not yet be done in previous research.
  • Moreover, chlorogenic acid is the compound having enhanced potential against gastric cancer, analyzed via HPLC.

BIOACTIVITIES

  • In Addition in-vivo trials of this plant are not yet performed.
  • Further the activities like anti-diabetic, anti-hemolytic, anti-pyretic, thrombolytic and hepatotoxicity analysis had not yet being interpreted, which are well described in this manuscript proving the enhanced potential of Thymus vulgaris.
  • The antioxidant analysis is being performed for the essential oil of Thymus vulgaris and not for simple plant extract of this plant.

COMPUTATIONAL ANALYSIS

  • Lastly, the computational analysis on inflammatory (COX-1 and COX-2) and gastric genes is not evaluated yet.
  • Furthermore, chlorogenic acid is the compound analyzed by HPLC analysis that has a neutralizing effect against gastric cancer, interpreted computationally via docking analysis of the particular compounds with gastric cancer causing genes.

Regards

Dr. Tariq Aziz (Postdoc, PhD)

Associate Professor

Reviewer 4 Report

The manuscript of Ayesha Saleem et al. is an interesting experiment showing the bioactive compounds of T. vulgaris, in-vitro and in vivo activities. My suggestions relate mainly to the GC-MS identification and presentation of results.

Title. Please use Thymus vulgaris.

Line 124. Please, the authors not used a derivation reagent (as silylation)? Was the extract solubilized and directly injected? How was made the identification, using only NIST? Please, the authors made kolvatz or retention index?

Line 391, table 1. Please add kolvatz or retention index in comparison with literature data.

Line 415, table 2. T. vulgaris italic. Please group only ethyl acetate and n-butanol with their respective compounds. Remove Sr. No column. In "fraction", keep ethyl acetate above, divide phytochemicals as phenolics above and flavonoids below. Create and divide in one other line ethyl acetate and butanol fraction. Likewise, make in butanol fraction. Please, add a column right side of phytochemicals and inform the r2 and standard curve. For example, shows y= ax+b, r2=xx. Remove the molecular weight of compounds with more than two decimal places.

Line 416, figure 2. Please use the chromatograms one above the other with their respective numeration.

Line 426, figure 3. Remove “administrator thursday, july 21, 2022… EA 28, EA 28”. Put cm-1 in the center. Use only the transmittance band number (ex 3339.77).

Author Response

Dear Ms.  Karol Zhang

Assistant Editor

Molecules

Please find the attached response to the reviewer comments on our article entitled “ HPLC and GC-MS Quantification of Phytoconstituents from T. vulgaris eliciting the potential of bioactive compounds by executing multiple in-vitro and in vivo biological activities inducing functionalized capabilities on COX-1, COX-2 and gastric cancer genes computationally” with manuscript ID molecules-1995076.

                                                                        Reviewer 4

Comments and Suggestions for Authors

The manuscript of Ayesha Saleem et al. is an interesting experiment showing the bioactive compounds of T. vulgaris, in-vitro and in vivo activities. My suggestions relate mainly to the GC-MS identification and presentation of results.

Title. Please use Thymus vulgaris.

AR: Thank You for your comment. Changes made in the title. [See title in the attached manuscript, highlighted in RED] in the revised manuscript.

Line 124. Please, the authors not used a derivation reagent (as silylation)? Was the extract solubilized and directly injected? How was made the identification, using only NIST? Please, the authors made kolvatz or retention index?

AR: Thank You for your comment. The Extract was solubilized and injected through intramuscularly route. The compounds identified via GC-MS analysis were further analyzed for the molecular weight, name, and chemical structure from NIST, a standard database which have all the details of the parameters regarding compound.

Line 391, table 1. Please add kolvatz or retention index in comparison with literature data.

AR: Thank You for your comment. Retention index added. [See Table 1: Column (Retention index) Highlighted in RED] in the revised manuscript.

Line 415, table 2. T. vulgaris italic. Please group only ethyl acetate and n-butanol with their respective compounds. Remove Sr. No column. In "fraction", keep ethyl acetate above, divide phytochemicals as phenolics above and flavonoids below. Create and divide in one other line ethyl acetate and butanol fraction. Likewise, make in butanol fraction. Please, add a column right side of phytochemicals and inform the r2 and standard curve. For example, shows y= ax+b, r2=xx. Remove the molecular weight of compounds with more than two decimal places.

AR: Thank You for your comment. All the instructions mentioned in above comment are being resolved. [See Table 2] in the revised manuscript.

Line 416, figure 2. Please use the chromatograms one above the other with their respective numeration.

AR: Thank You for your comment. By pacing chromatograms one above the other were causing some issue. The Numeration A, B, C and D are determining the sequence of chromatograms in the revised manuscript.

Line 426, figure 3. Remove “administrator thursday, july 21, 2022… EA 28, EA 28”. Put cm-1 in the center. Use only the transmittance band number (ex 3339.77).

AR: Thank You for your comment. All changes have been made in the revised manuscript.

Regards

Dr. Tariq Aziz (Postdoc, PhD)

Associate Professor

Round 2

Reviewer 1 Report

Authors have tried to revise/explain the recommended items, nonetheless, based on my previous reasons I am still not satisfied with the novelty and aim of the study.

Author Response

Reviewer 1 (Round 2)

Authors have tried to revise/explain the recommended items, nonetheless, based on my previous reasons I am still not satisfied with the novelty and aim of the study.

AR: Thank you so much for your comments. The articles shared by the reviewer are appreciable, but the information shared by the perspective reviewer lacks the points of novelty mentioned by us. We do agree that Thymus vulgaris is a well-known plant and a lot of analysis has been performed on it, but the some of the bioactivities and especially computational analysis provided in our manuscript are not performed previously. Yet, we believe that all the information provided in the manuscript will prove to be a benchmark for all the scientific society as this paper describes all possible aspects on well-known Thymus vulgaris plant.

Regards

Dr. Tariq Aziz (Postdoc, PhD)

Professor (Associate)

School of Food & Biological Engineering

Jiangsu University, Zhenjiang, 212013, China

Reviewer 3 Report

The reviewers have attempted to answer my question, I appreciate the efforts. Either the authors are lying or they do not know about the extensive literature already out there on thymus Vulgaris biochemical analysis and evaluation. Below is my reply on why this work is still not suitbale for publication.

GC-MS AND HPLC ANALYSIS

The fractions of n-Butanol and Ethyl Acetate of this plant had not been identified previously via GC-MS and HPLC quantification.

This statement is incorrect, please see below: https://www.ncbi.nlm.nih.gov/pmc/articles/PMC3609397/

FTIR analysis of functional groups interpretation of Thymus vulgaris has not yet be done in previous research.

This is also incorrect, plus FTIR analysis is not confirmatory, and tons of peaks remain unaccounted for. Multidimensional NMR, X-ray, and High-resolution mass spectrometry data are used for proper characterization. Please see below for FTIR analysis of the Thymus Vulgaris essential oil.

https://www.tandfonline.com/doi/abs/10.1080/10412905.2017.1351405?journalCode=tjeo20

Moreover, chlorogenic acid is the compound having enhanced potential against gastric cancer, analyzed via HPLC.

This is not new either. Chlorogenic acid has been analyzed via HPLC long ago: 

https://www.scielo.br/j/bjps/a/CQWw4x6xHwYPbfrpcyRrbKx/?format=pdf&lang=en

https://www.sciencedirect.com/science/article/abs/pii/S0926669012004992

BIOACTIVITIES

In Addition in-vivo trials of this plant are not yet performed.

I am sorry, what does this statement even mean? Both in vitro and in vivo studies have been performed previously.

https://www.ncbi.nlm.nih.gov/pmc/articles/PMC6606503/

https://www.mdpi.com/1422-0067/20/7/1749/htm

Further the activities like anti-diabetic, anti-hemolytic, anti-pyretic, thrombolytic and hepatotoxicity analysis had not yet being interpreted, which are well described in this manuscript proving the enhanced potential of Thymus vulgaris.

What do the authors mean by " not being interpreted"? There is tons of research out there on this topic already.

https://www.ncbi.nlm.nih.gov/pmc/articles/PMC7501980/

The antioxidant analysis is being performed for the essential oil of Thymus vulgaris and not for simple plant extract of this plant.

Another false statement:

https://www.scirp.org/journal/paperinformation.aspx?paperid=33090

COMPUTATIONAL ANALYSIS

Lastly, the computational analysis on inflammatory (COX-1 and COX-2) and gastric genes is not evaluated yet.

Yet again false statement:

https://www.ncbi.nlm.nih.gov/pmc/articles/PMC4524694/

Furthermore, chlorogenic acid is the compound analyzed by HPLC analysis that has a neutralizing effect against gastric cancer, interpreted computationally via docking analysis of the particular compounds with gastric cancer causing genes.
How do authors know if chlorogenic acid makes it all the way to the gastric cancer-causing genes and affects their activity? This analysis is pure speculative at best.

Author Response

Reviewer 2 (Round 2)

The reviewers have attempted to answer my question, I appreciate the efforts. Either the authors are lying, or they do not know about the extensive literature already out there on thymus Vulgaris biochemical analysis and evaluation. Below is my reply on why this work is still not suitbale for publication.

AR: Thank you so much for your comments. The articles shared by the reviewer are appreciable, but the information shared by the perspective reviewer lacks the points of novelty mentioned by us. We do agree that Thymus vulgaris is a well-known plant and a lot of analysis has been performed on it, but the some of the bio-activities and specially computational analysis provided in our manuscript are not performed previously. Yet, we believe that all the information provided in the manuscript will prove to be a benchmark for all the scientific society as this paper describes all possible biological aspects on well-known Thymus vulgaris plant.

Below is the clarification of our work and work done in previous studies attached nicely by the reviewer.

GC-MS AND HPLC ANALYSIS

The fractions of n-Butanol and Ethyl Acetate of this plant had not been identified previously via GC-MS and HPLC quantification.

This statement is incorrect, please see below: https://www.ncbi.nlm.nih.gov/pmc/articles/PMC3609397/

 AR: Thank you for sharing the article links. We do agree with these fruitful studies, but in our current study the compounds identified through these fractions are different from the phytoconstituents identified previously.

FTIR analysis of functional groups interpretation of Thymus vulgaris has not yet be done in previous research.

This is also incorrect, plus FTIR analysis is not confirmatory, and tons of peaks remain unaccounted for. Multidimensional NMR, X-ray, and High-resolution mass spectrometry data are used for proper characterization. Please see below for FTIR analysis of the Thymus Vulgaris essential oil.

https://www.tandfonline.com/doi/abs/10.1080/10412905.2017.1351405?journalCode=tjeo20

 AR: Thank you for sharing the article link. This study was conducted on essential oil of the particular plant and not on the simple plant tincture.

Moreover, chlorogenic acid is the compound having enhanced potential against gastric cancer, analyzed via HPLC.

This is not new either. Chlorogenic acid has been analyzed via HPLC long ago: 

https://www.scielo.br/j/bjps/a/CQWw4x6xHwYPbfrpcyRrbKx/?format=pdf&lang=en

https://www.sciencedirect.com/science/article/abs/pii/S0926669012004992

 AR: Thank you for sharing the article link. [The highest contents of chlorogenic acid, rosmarinic acid, and caffeic acid were found in Lonicera japonica flowering buds, Melissa officinalis leaves, and Coffea canephora seeds at the concentration of 9.900 ± 0.004, 19.908 ± 0.171, and 1.233 ± 0.003 g/100 g of dried plant, respectively]. These lines are from the article attached by the reviewer, which does not report the high concentration of chlorogenic acid in the attached manuscripts. In one the manuscript only the Thymus vulgaris is being used from which chlorogenic acid is being identified yet its enhanced properties are not being discussed till now.

BIOACTIVITIES

In Addition in-vivo trials of this plant are not yet performed.

I am sorry, what does this statement even mean? Both in vitro and in vivo studies have been performed previously.

https://www.ncbi.nlm.nih.gov/pmc/articles/PMC6606503/

https://www.mdpi.com/1422-0067/20/7/1749/htm

  AR: Thank you for sharing the article link. We have gone through these articles as well. In these articles the in vitro and in vivo studies mentioned are different in all aspects of our manuscripts. We have performed in-vivo anti-inflammatory and hepatoprotective activities where as in these articles in-vivo analysis are performed on breast cancer cell lines and histone modification analysis, thus our in-vivo trials are totally different from the previous in-vivo analysis.

Further the activities like anti-diabetic, anti-hemolytic, anti-pyretic, thrombolytic and hepatotoxicity analysis had not yet being interpreted, which are well described in this manuscript proving the enhanced potential of Thymus vulgaris.

What do the authors mean by " not being interpreted"? There is tons of research out there on this topic already.

https://www.ncbi.nlm.nih.gov/pmc/articles/PMC7501980/

  AR: Thank you for sharing the article link. Dear reviewer, in the above mentioned articles only ant-diabetic activity has been discussed and not anti-hemolytic, anti-pyretic, thrombolytic and hepatotoxicity analysis, which are performed in this manuscript study. Yet, for antidiabetic activity the assays used in our study are totally different from the previously reported analysis.

The antioxidant analysis is being performed for the essential oil of Thymus vulgaris and not for simple plant extract of this plant.

Another false statement:

https://www.scirp.org/journal/paperinformation.aspx?paperid=33090

 AR: Thank you for sharing the article link. [Essential oil of Thymus vulgaris L. and Rosmarinus officinalis L.: Gas chromatography-mass spectrometry analysis, cytotoxicity and antioxidant properties and antibacterial activities against foodborne pathogens]. This is the title of the article link attached by you dear reviewer, in which it is clearly mentioned that Essential oil of Thymus vulgaris, whereas we have conducted our study on only simple plant tincture. Yet, the assays used for anti-oxidant activity are totally differently performed in our manuscript study.

COMPUTATIONAL ANALYSIS

Lastly, the computational analysis on inflammatory (COX-1 and COX-2) and gastric genes is not evaluated yet.

Yet again false statement:

https://www.ncbi.nlm.nih.gov/pmc/articles/PMC4524694/

  AR: Thank you for sharing the article link. We have gone through these articles, yet the our computationally analysis is again novel we have analyzed thymol and chlorogenic acid against COX-1, COX-2, and gastric cancer genes, respectively, proving to be the most potent phytochemicals of Thymus vulgaris. Previously, none of the phytochemical of this plant has been analyzed as potential compound, rather the above mentioned articles shows the potential of Naproxen, Etoricoxib, Flurbiprofen, Ibuprofen, Indomethacin, Ketoprofen, Piroxicam, Diclofinac, Ketorolac, Tolmetin, Tenoxicam, Valdecoxib, Meloxica, Phenylbutazone, Rofecoxib, Sulindac, Celecoxib drugs against only anti-inflammatory genes.

Furthermore, chlorogenic acid is the compound analyzed by HPLC analysis that has a neutralizing effect against gastric cancer, interpreted computationally via docking analysis of the particular compounds with gastric cancer causing genes.
How do authors know if chlorogenic acid makes it all the way to the gastric cancer-causing genes and affects their activity? This analysis is pure speculative at best.

AR: Thank You so much for your comment. Dear reviewer, we have analyzed all the compounds obtained from HPLC analysis computationally of which the chlorogenic acid showed the highest binding affinity with the gastric cancer causing genes, which shows the remarkable potential of chlorogenic acid against gastric cancer.

Regards

Dr. Tariq Aziz (Postdoc, PhD)

Professor (Associate)

School of Food & Biological Engineering

Jiangsu University, Zhenjiang, 212013, China

Reviewer 4 Report

The authors made all corrections and are ready for publication.

Author Response

Reviewer 4 (Round 2)

The authors made all corrections and are ready for publication.

AR: Thank You so much for your comment. It means a lot to us. Thank You!

Regards

Dr. Tariq Aziz (Postdoc, PhD)

Professor (Associate)

School of Food & Biological Engineering

Jiangsu University, Zhenjiang, 212013, China
